# Efficient use of a Lagrangian Particle Dispersion Model for atmospheric inversions using satellite observations of column mixing ratios

Rona L. Thompson[1], Nalini Krishnankutty[1], Ignacio Pisso[1], Philipp Schneider[1], Kerstin Stebel[1], Motoki Sasakawa[2], Andreas Stohl[3] and Stephen Platt[1]

[1]NILU, Kjeller, Norway
[2]Earth System Division, National Institute for Environmental Studies, Tsukuba, Japan
[3]University of Vienna, Vienna, Austria

*Correspondence to*: Rona L. Thompson (rlt@nilu.no)

## Abstract

Satellite instruments for measuring atmospheric column mixing ratios have improved significantly over the past couple of decades with increases in pixel resolution and accuracy. As a result, satellite observations are being increasingly used in atmospheric inversions to improve estimates of emissions of greenhouse gases (GHGs), particularly $CO_2$ and $CH_4$, and to constrain regional and national emission budgets. However, in order to make use of the increasing resolution in inversions, the atmospheric transport models used need to be able to represent the observations at these finer resolutions. Here, we present a new and computationally efficient methodology to model satellite column average mixing ratios with a Lagrangian Particle Dispersion Model (LPDM) and calculate the Jacobian matrices describing the relationship between surface fluxes of GHGs and atmospheric column average mixing ratios, as needed in inversions. The development will enable a more accurate representation of satellite observations (especially high-resolution ones) through using LPDMs and thus help to improve the accuracy of emissions estimates obtained by atmospheric inversions. We present a case study using this methodology in the LPDM, FLEXPART, and the inversion framework, FLEXINVERT, to estimate $CH_4$ fluxes over Siberia using column average mixing ratios of $CH_4$ ($XCH_4$) from the TROPOMI instrument onboard the Sentinel-5P satellite. The results of the inversion using TROPOMI $XCH_4$ are evaluated against results using ground-based observations.

## 1. Introduction

Satellite remote sensing provides a wealth of information on the atmosphere and its composition. The number of satellite missions monitoring long-lived greenhouse gases (GHGs), specifically $CO_2$ and $CH_4$, has grown substantially over the past couple of decades providing information on their variability, trends, and sources. From instruments onboard satellites it is possible to retrieve mixing ratios of GHGs, most commonly as column averages (e.g. $XCO_2$ and $XCH_4$) or, depending on the instrument, viewing angle and retrieval, as sub-columns, which can be used to derive estimates of surface-atmosphere fluxes

using atmospheric transport models and inversion techniques (e.g. Alexe et al., 2015; Chen et al., 2023; Chevallier et al., 2005; Peiro et al., 2022; Zhang et al., 2023, 2021). Satellite instruments can be classified as "area flux mappers" or "point source imagers" (Jacob et al., 2022). Area flux mappers are high precision instruments with larger pixel sizes (order of 0.1 to 10 km) and are designed to image mixing ratios on regional to global scales while point source imagers have smaller pixel size (< 0.1

km) and are designed to detect and quantify individual sources by mapping their plumes (Jacob et al., 2022). Specifically, it is the area flux mappers that are useful in inversions to estimate global and regional fluxes of $CO_2$ and $CH_4$, and they are becoming increasingly used to determine regional and national emission budgets and for comparisons with GHG emission inventories (e.g. Byrne et al., 2023; Deng et al., 2022; Maasakkers et al., 2021; Nesser et al., 2023; Worden et al., 2022). As satellite instrumentation has improved, there has also been an increase in pixel resolution. For example, for $CH_4$ the earliest satellite

observations were from the SCIAMACHY instrument onboard ENVISAT, launched in 2002, which had a pixel size of 30 × 60 km, while the $XCH_4$ product from TROPOMI (TROPOspheric Monitoring Instrument) onboard Sentinel-5P, launched in 2017, has a pixel size at nadir of 5.5 × 7 km, and the recently launched MethaneSAT has a pixel size of just 0.1 × 0.4 km (Jacob et al., 2022). With this resolution increase it is important to consider how the observations are represented in atmospheric transport models for inverse modelling.


Up to the present, inversions with satellite observations have primarily been made using Eulerian atmospheric transport models, either using Green's functions, adjoint models or ensemble approaches to relate the column average mixing ratios to fluxes (e.g. Bergamaschi et al., 2009; Tsuruta et al., 2023; Varon et al., 2023). This is because the number of model iterations or ensemble members is independent of the number of observations, which for satellites can be a very large number. In contrast,

there are only very few examples in the literature in which Lagrangian particle dispersion models (LPDMs) have been used with satellite observations (e.g. Ganesan et al. 2017; Wu et al. 2018) because the number of model calculations required in their backwards mode is proportional to the number of observations making it computationally demanding.

On the other hand, LPDMs have some advantages over Eulerian models. LPDMs exhibit less numerical diffusion compared

to Eulerian models, and because of this, generally better capture tracer filaments generated by atmospheric dispersion (Ottino, 1989) and fine structures in tracers mixing ratios resulting from long-range transport (Rastigejev et al., 2010). Furthermore, LPDMs can accurately represent any observation geometry, whereas in Eulerian models an observation is represented by a grid cell (Pisso et al., 2019). With the increasing resolution of satellite instruments, Eulerian model resolution may become a limiting factor in the ability to accurately represent an observation, hence the use of LPDMs becomes a more interesting

alternative. Finally, LPDMs can be run in a backwards in time mode (without significant modifications to the code), which allows the sensitivity of an observation to fluxes to be calculated, and in this way LPDMs are sometimes said to be "self adjoint".

The main challenge of using LPDMs with satellite observations is the number of observations for which backwards

computations need to be made, and that each retrieval is made over numerous vertical layers, which need to be represented in the model and combined with the retrieval's averaging kernel and prior mixing ratio profile. Here, we present a novel methodology for representing total column observations efficiently in an LPDM and open the possibility to use LPDMs across a variety of spatial and temporal scales for atmospheric inversions with satellite data. The methodology has been implemented in the LPDM, FLEXPART (FLEXible PARTicle dispersion model, version 10.4, Pisso et al. 2019), but could be in principle

be implemented in any LPDM. It can also be applied to any satellite retrieval.

We demonstrate the use of this methodology in a case study looking at $CH_4$ using retrievals from TROPOMI on board the Sentinel-5P satellite. The case study region is Siberia, which was chosen because it has significant $CH_4$ emissions from both natural sources (e.g. peatlands) and anthropogenic sources (e.g., fugitive emissions from oil and gas extraction and

transportation as well as from coal mining). We focus on the year 2020 and for the months March to October when there are an appreciable number of retrievals available. We use FLEXPART to model $XCH_4$ and, along with observed $XCH_4$, we derive estimates of $CH_4$ fluxes using the Bayesian inversion framework, FLEXINVERT (Thompson and Stohl, 2014). The fluxes from inversions using TROPOMI $XCH_4$ are evaluated by comparing with fluxes derived from inversions using ground-based observations from the JR-STATION in-situ measurement network (Sasakawa et al., 2010, 2012), plus one in-situ site and two

flask sampling sites available form the World Data Centre for Greenhouse Gases (WDCGG).

## 2. Methodology

FLEXPART models atmospheric transport using virtual particles that are subject to transport and turbulent mixing as determined from meteorological fields. FLEXPART can be run in a backwards in time mode, which is theoretically consistent with the forward time mode calculations (Flesch et al., 1995; Thomson, 1990), to calculate the residence time of virtual

particles in a surface layer within the boundary layer and thus the influence of surface fluxes on these particles. In this mode, Jacobian matrices representing the influence of fluxes on an atmospheric observation can be derived, which are termed "source-receptor-relationships" (SRRs), or sometimes "footprints" (Seibert and Frank, 2004). The SRRs can be integrated with flux fields to simulate mixing ratios and can be used in atmospheric inversions to update a prior estimate of the fluxes (e.g. Thompson and Stohl, 2014; Brioude et al., 2012). However, as far as we are aware, LPDMs have not been used before to

model large numbers of satellite observations owing to the computational cost.

### 2.1 Modelling total column observations

Following Rodgers and Connor (2003) a modelled vertical profile, $\mathbf{x}$ of mixing ratios can be compared with a retrieved profile using:

$$\mathbf{x}^{sm} = \mathbf{x}^{pri} + \mathbf{A}(\mathbf{x} - \mathbf{x}^{pri}) \tag{1}$$

where $\mathbf{A}$ is the averaging kernel matrix (an $N \times N$ dimensional matrix), $\mathbf{x}^{pri}$ is the a priori profile used in the retrieval and $\mathbf{x}^{sm}$ is a smoothed version of the vertical profile, $\mathbf{x}$. For a retrieval performed on $N$ discrete pressure layers, the column average mixing ratio, $x^{avg}$ is the weighted sum of $N$ sub-columns corresponding to the retrieval pressure layers (Apituley et al., 2021):

$$x^{avg} = x^{avg,pri} + \sum_{n=1}^{N} a_n w_n x_n - \sum_{n=1}^{N} a_n w_n x_n^{pri} \tag{2}$$

where $x^{avg,pri}$ is the prior column average mixing ratio, $a_n$ is the $n$th element of the column averaging kernel, $w_n$ is a pressure weighting term related to the thickness of the pressure layer, and $x_n$ is the modelled mixing ratio for the $n$th retrieval layer. For LPDMs, $x_n$ can be modelled as the product of the SRR for the $n$th retrieval layer, $\mathbf{H}_n$, (a $1 \times q$ dimensional matrix, where $q$ is the number of flux variables) and the estimate of the fluxes, $\mathbf{f}$, $(q \times 1)$, plus an estimate of the so-called "background" mixing ratio. Following Thompson and Stohl (2014), the background mixing ratio is modelled as the product of the background-receptor-relationship (BRR) matrix calculated from the positions of the virtual particles when they terminate, $\mathbf{H}_n^{ini}$ and a 3D field of initial mixing ratios, $\mathbf{y}^{ini}$. Note, that chemical losses during the backward simulation period, e.g., from OH oxidation, can be taken into account in the SRR and BRR matrices. The modelled mixing ratio for the $n$th retrieval layer is thus:

$$x_n = \mathbf{H}_n \mathbf{f} + \mathbf{H}_n^{ini} \mathbf{y}^{ini} \tag{3}$$

By substituting Eq. 3 into Eq. 2 we obtain:

$$x^{avg} = x^{avg,pri} + \sum_{n=1}^{N} a_n (\mathbf{H}_n \mathbf{f} w_n + \mathbf{H}_n^{ini} \mathbf{y}^{ini} w_n) - \sum_{n=1}^{N} a_n w_n x_n^{pri} \tag{4}$$

Thus, the column average SRR can be expressed as:

$$\mathbf{H}^{col} = \sum_{n=1}^{N} a_n \mathbf{H}_n w_n \tag{5}$$

and similarly for the column average BRR, $\mathbf{H}^{col,ini}$.

Equations 2 to 5 represent the approach that has previously been used to model satellite observations with an LPDM, namely, the mixing ratio, $x_n$ has been calculated for each retrieval layer requiring calculation of the SRR, $\mathbf{H}_n$ for each layer (Ganesan et al. 2017; Wu et al. 2018). Equation 5 requires maintaining information about each retrieval layer in order to calculate the column SRR. Our method (as described below) departs from this approach and is much more computationally efficient.

In FLEXPART (and LPDMs generally), SRRs are calculated by sampling the particles on a regular three-dimensional grid. In general, for a grid cell, $i$, the SRR for retrieval layer $n$ is calculated as:

$$\mathbf{H}_{in} = \frac{1}{P_n} \sum_{p=1} \frac{l_{pin} \Delta t_{pin}}{\rho_i} \tag{6}$$

where $l_{pin}$ is the transmission function for particle $p$ (and represents the fraction of the mass remaining in the particle, which can change after release in the case of atmospheric chemistry), $\Delta t_{pin}$ is the residence time of the particle in the grid cell, $\rho_i$ is the air density in the grid cell and $P_n$ is the number of particles released in layer $n$ (Seibert and Frank, 2004). (Note in Eq. 6 the number of particles summed over is not specified as this depends on the number of particles that reside in the grid cell $i$ and is $\leq P_n$). Thus the column SRR relationship is found by substituting Eq. 6 into Eq. 5:

$$\mathbf{H}_i^{col} = \frac{1}{\rho_i} \sum_{n=1}^{N} a_n w_n \frac{1}{P_n} \sum_{p=1} l_{pin} \Delta t_{pin} \tag{7}$$

However, using Eq. 7 would still require maintaining information about which retrieval layer a particle had originated from in order to calculate the column SRR and, furthermore, if $P_n$ has the same value for all layers, this would mean that all layers are sampled equally even though particles originating in upper layers are much more unlikely to reach the surface layer and thus contribute to the SRR.

Instead, we carry the information of $a_n w_n$ by varying the particle density in each layer, where the number of particles released per layer, $P_n$, is:

$$P_n = P a_n w_n \tag{8}$$

where $P$ is the total number of particles released per retrieval. By substituting $a_n w_n$ for $P_n/P$ into Eq. 7 we derive:

$$\mathbf{H}_i^{col} = \frac{1}{\rho_i} \frac{1}{P} \sum_{n=1}^{N} \sum_{p=1} l_{pin} \Delta t_{pin} \tag{9}$$

Equation 9 can be simplified further by noting that the sum over $N$ layers and the sum over particles in the grid cell $i$ originating from each layer is equivalent to summing over the particles in the grid cell originating from all layers (hence we omit the index $n$):

$$\mathbf{H}_i^{col} = \frac{1}{\rho_i} \frac{1}{P} \sum_{p=1} l_{pi} \Delta t_{pi} \tag{10}$$

In Eq. 10 the information on the retrieval layer from which a particle originated from does not need to be kept (as it is taken into account at the particle initialization via $P_n$) and the equation is analogous to that for point observations. This implementation was compared to calculating the SRRs for each layer individually, and the results were the same within the limits of numerical rounding errors. However, by implementing the calculation this way, total column observations can be simulated with the same computational cost as for point observations.

Similarly, the total column BRR for the grid cell $i$ can be obtained as:

$$\mathbf{H}_i^{col,ini} = \frac{1}{P} \sum_{p=1} l_{pi} \tag{11}$$

A further consideration when modelling satellite observations with a Lagrangian model, is the geometry of the retrieval, since the ground-based pixels are not necessarily rectangular and can be rotated with respect to the meridians and parallels. In FLEXPART (and LPDMs generally), an observation is represented by releasing virtual particles from a volume in which the particles are distributed randomly. However, the default is that this volume is rectangular and aligned with the meridians and parallels. Therefore, we have implemented an affine transformation on the particle positions so that the volume they represent matches the geometry of the retrieval (see the supplementary information for a complete description of the affine algorithm).

## 2.2 Averaging of retrievals

Even with the efficient modelling of total column measurements using the method described above, current satellite missions can provide on the order of 10,000 to 100,000 retrievals globally per day, making the cost of computing backward trajectories for each retrieval still computationally expensive if the study region is large. For this reason, we average the retrievals to so-

called "super observations". Averaging retrievals also has the advantage that the random error of the "super observation" is

160 reduced compared to each individual retrieval.

However, in some areas where there is strong heterogeneity in the column average measurements, for instance, due to large localized sources, it would be advantageous to keep higher resolution observations. Therefore, we developed an optimal averaging routine in which the degree of averaging is based on the standard deviation of the column average mixing ratios.

The retrievals are averaged to rectangular grid cells that are aligned with the meridians and parallels. The user decides the finest resolution grid cell to be used, $d_{min}$, and the number of resolution steps, *nsteps*. The averaging is first performed for the coarsest resolution (given as $d_{min} \times 2^{nsteps-1}$), then the averaging is refined stepwise (from *nsteps*–1 to 0) by dividing grid cells into four where the standard deviation (recalculated at each step) is above a given threshold. If there are any quarters of the grid cell (as defined for the current step) where there are no retrievals, then the grid cell is any how divided in the next

iteration to avoid having a super-observation for a grid cell that is not fully represented by the retrievals. Retrievals which are outside ±2 standard deviations of the grid cell mean are not included in the average to avoid the influence of large outliers. The averaging is redefined each day based on the available retrievals.

In the algorithm, the column average mixing ratio corresponding to the average of *M* retrievals is calculated as:

$$\bar{x} = \frac{1}{S}\sum_{m=1}^{M}\left(s_m x_m^{pri} + \sum_{n=1}^{N} s_m a_{mn} w_{mn}\left(x_{mn} - x_{mn}^{pri}\right)\right) \tag{12}$$

where $\bar{x}$ is the total column mixing ratio corresponding to the average, $x_m^{pri}$ is the prior column average mixing ratio of the *m*th retrieval, *N* is the number of vertical layers in the retrievals, $s_m$ is the surface area, and *S* is the total surface area of all retrievals. By rearranging Eq. 12 we obtain:

$$\bar{x} = \frac{1}{S}\sum_{m=1}^{M} s_m x_m^{pri} + \frac{1}{S}\sum_{m=1}^{M}\sum_{n=1}^{N} s_m a_{mn} w_{mn} x_{mn} - \frac{1}{S}\sum_{m=1}^{M}\sum_{n=1}^{N} s_m a_{mn} w_{mn} x_{mn}^{pri} \tag{13}$$

and this can be written as:

$$\bar{x} = \overline{x^{pri}} + \sum_{n=1}^{N} \overline{x_n}\, \overline{a_n w_n} - \frac{1}{S}\sum_{m=1}^{M}\sum_{n=1}^{N} s_m a_{mn} w_{mn} x_{mn}^{pri} \quad (if\ all\ x_{mn} = \overline{x_n}) \tag{14}$$

where $\overline{x^{pri}}$ is the area-weighted-average prior column mixing ratio, and $\overline{a_n w_n}$, $\overline{x_n}$ are the area-weighted-average column averaging kernel and pressure weighting, and mixing ratio corresponding to the *n*th layer. Note that the condition *if all* $x_{mn} = \overline{x_n}$ is met when the particle release is made for the area over which the *M* retrievals are averaged. The uncertainty of the super

observation is calculated as the quadratic sum of the uncertainties of the individual retrievals weighted by the ground-pixel area of the retrievals.

For FLEXPART users, we include a description of the changes to the v10.4 code for the implementation of this methodology in the Supplement. In addition, we include a brief description of how these developments could be used with the recently

released FLEXPART v11.

## 3. Case study on methane sources in Siberia

We demonstrate the use of FLEXPART for modelling column average mixing ratios, as well as the averaging algorithm, in a case study looking at $CH_4$ emissions in Siberia. Siberia was chosen as it is a region with significant $CH_4$ emissions from both natural sources (e.g. peatlands) and anthropogenic sources (e.g., fugitive emissions from oil and gas extraction and transportation as well as from coal mining). It was also in Siberia that one of the largest point source emissions of $CH_4$ was detected by GHGSat – from a coal mine in the Kemerovo region (https://www.bbc.com/news/science-environment-61811481) and significant $CH_4$ sources have also been detected in this region by TROPOMI (Trenchev et al., 2023). On the other hand, Siberia is a challenging region to study, since there are no observations available north of around 50°N in the winter and the region is often cloudy, further limiting the number of available retrievals. Our inversion domain covers western and central Siberia (50° to 115°E and 40° to 80°N) and includes the Western Siberian Lowlands and major oil and gas fields, as well as important coal mining regions such as Kemerovo (Fig. 1). We limit the period of our study to March to October – months during which there are observations also north of 50°N and focus on the year 2020.

To evaluate the inversions using TROPOMI XCH4, we make use of the JR-STATION ground-based network of $CH_4$ measurements. JR-STATION consists of 9 sites with in-situ measurements of $CH_4$ with the first sites having data from 2004 (Sasakawa et al., 2010, 2012).

### 3.1. Case study methodology

### 3.1.1 Inversion method

For the inversion we use the FLEXINVERT Bayesian inversion framework as described by Thompson and Stohl (2014). In this framework, the optimal fluxes are those that minimise the cost function:

$$J(\mathbf{z}) = \frac{1}{2}(\mathbf{z} - \mathbf{z}_b)^{\mathrm{T}}\mathbf{B}^{-1}(\mathbf{z} - \mathbf{z}_b) + \frac{1}{2}(\mathbf{Hz} - \mathbf{y})^{\mathrm{T}}\mathbf{R}^{-1}(\mathbf{Hz} - \mathbf{y}) \qquad (15)$$

where $\mathbf{B}$ is the prior error covariance matrix and describes the error and error correlation of the prior fluxes, $\mathbf{R}$ is the observation error covariance matrix and describes the uncertainty in the observations, $\mathbf{z}_b$ is the prior state vector, $\mathbf{z}$ is the optimal (or posterior) state vector, and $\mathbf{y}$ is the observation vector. The minimum of the cost function is found using the M1QN3 quasi-Newton algorithm (Gilbert and Lemaréchal, 1989). This algorithm does not provide an estimate of the Hessian matrix $\nabla^2 J(\mathbf{z})$, therefore, the posterior uncertainty was calculated instead using a Monte Carlo ensemble following Chevallier et al. (2007).

The state vector variables include offsets to the prior fluxes, which are resolved at 14-day temporal resolution and at varying spatial resolution from 0.5° to 2.0° depending on how strongly the fluxes influence the observations (Thompson and Stohl, 2014). The spatial resolution of the state vector was calculated separately for the TROPOMI and for the ground-based observations (see Supplementary Fig. S1 for maps of the spatial grid). For each 14-day interval this resulted in 2619 flux variables for the TROPOMI and 5920 for the ground-based observation inversions. Note, only land fluxes were optimized in

the inversions, since the observations have little sensitivity to fluxes over the ocean. In any case, the contribution from ocean fluxes on the modelled mixing ratios was accounted for (only these fluxes were fixed to the prior values and not optimized).

The state vector also includes scalars of the boundary conditions (i.e., the initial mixing ratios in 3D space represented by the vector, $\mathbf{y}^{ini}$ in Eq. 3). The scalars are defined for four latitudinal bands, 90°-30°N, 30°N-0°, 0°-30°S, and 30°-90°S, and for three vertical layers, from 0-2000, 2000-10,000 and 10,000-70,000 metres above ground level, and are optimized for 28-day averages. The uncertainty in the scalars was set at 5% for the TROPOMI inversions and at 1% for the ground-based observation inversions. A larger uncertainty was chosen for the TROPOMI inversions after initial tests showed that lower uncertainties did

not allow sufficient freedom to correct erroneous prior background mixing ratios (see Section 3.2.2).

For the case study, FLEXPART was run using the ECMWF meteorological reanalysis data ERA5 at 0.5° × 0.5° and hourly resolution. Backwards trajectories were made using 30,000 particles for each TROPOMI super-observation. The trajectories were calculated for 20 days backwards in time from the time of the observation. The SRRs were calculated at 0.5° over the

235 inversion domain and at 2.0° globally. In addition, the BRR ($\mathbf{H}_n^{ini}$ in Eq. 3) was calculated at the termination of the particles. For comparison with the inversions using TROPOMI retrievals, we also performed inversions using ground-based observations. The FLEXPART runs for these observations used the same set-up as for the retrievals but with only 20,000 particles per observation, which was deemed sufficient to represent a point observation.

The so-called background mixing ratio for each column average observation, was calculated as $\mathbf{H}^{col,ini}\mathbf{y}^{ini}$ (see Eq. 3-5) where $\mathbf{y}^{ini}$ is a 3D field of $CH_4$ mixing ratios (resolved daily) and was taken from the CAMS data assimilation product, EGG4 (https://ads.atmosphere.copernicus.eu/datasets/cams-global-ghg-reanalysis-egg4?tab=overview). In addition, we have used 3D mixing ratio fields from the CAMS Greenhouse gas inversion product TM5-4DVAR (https://atmosphere.copernicus.eu/greenhouse-gases-supplementary-products) in a sensitivity test.

**3.1.2 Observations**

In this case study, we use the Weighting Function Modified Differential Optical Absorption Spectroscopy (WFMD) retrieval product (version 1.8) from the University of Bremen (Schneising et al., 2019, 2023). We selected retrievals that had a quality flag of 0 (where the quality value is either 0 (good) or 1 (bad)). The retrievals were averaged to super observations, as described in Sect. 2.2, using two resolution steps with grid cell sizes of 0.25° and 0.5°. Uncertainties for the super observations were

250 calculated as the quadratic sum of the uncertainty for each retrieval weighted by the area of the ground-pixel of the retrieval. The full observation-space uncertainty was the quadratic sum of the super-observation uncertainty and an uncertainty estimated for the background column average mixing ratio. The resulting observation space uncertainties were typically in the range of 14 to 20 ppb with a median value of 16 ppb. The square of these uncertainties were used as the variances in the observation error covariance matrix, and we assumed that errors in the super observations were uncorrelated. On average, there were 3781

super-observations per day (see Supplementary Fig. S2 for an overview of the number of super-observations by latitude and state vector time step).

For validation, we used ground-based observations from the Japan-Russia Siberian Tall Tower Inland Observation Network (JR-STATION) network (Sasakawa et al., 2010, 2012; 2025) (Fig. 1 and Table 1). It is comprised of nine (currently six operating) tower sites in Siberia where simultaneous multi-point semi-continuous observations of $CO_2$ and $CH_4$ have been made. The $CH_4$ mixing ratios were measured using a modified $SnO_2$ semiconductor sensor and determined against the NIES 94 $CH_4$ scale. The NIES 94 $CH_4$ scale ranges approximately 5 ppb higher than the WMO-CH4-X2004A scale, thus we adjusted the $CH_4$ mixing ratios to the WMO scale for use in the inversions. The JR-STATION system measures the air at each intake height on the tower for three minutes at a time before switching the airflow to the next height. The 3-minute values for each height are averaged to obtain a representative value for each hour. In addition, we used observations from the flask sampling sites Ulaan Uul, Mongolia (UUM) and Teriberka, Russia (TER), and in-situ observations from the Global Atmospheric Watch site, Cholpon-Ata, Kyrgyzstan (CPA), which were all obtained from the World Data Centre for Greenhouse Gases (WDCGG). These data were filtered to remove observations that were flagged as "invalid" and for the flask data, pairs of flasks were averaged to one observation. The observation space uncertainties were typically in the range of 11 to 14 ppb with a median value of 12 ppb.

### 3.1.3 Prior information

In the case study inversion, we optimize the total net $CH_4$ flux. A prior estimate for the total net flux was prepared using the following input datasets: i) the EDGAR-v8 for anthropogenic emissions (Crippa et al., 2023), ii) the land-surface model, LPX-Bern for natural fluxes from peatlands, wet and inundated soils, and the soil sink, iii) Etiope et al. (2019) for geological emissions, iv) GFED-v4.1s for biomass burning emissions (Werf et al., 2017), and v) the observation-based climatology of Weber et al. (2019) for ocean fluxes. An overview of the flux estimates used in the prior is given in Table 2. The input data are given at different temporal and spatial resolutions and thus were averaged/interpolated to the same spatial resolution as the SRRs for the inversion domain, i.e., 0.5° and interpolated to 14 days to match the state vector temporal resolution.

For the inversion using ground-based observations, prior uncertainties were calculated for each grid cell as 50% of the prior estimate but with a lower limit of $1\times10^{-9}$ kg m$^{-2}$ h$^{-1}$, which is approximately the 10th percentile value of all fluxes over the inversion domain. For the inversions using TROPOMI, after a first inversion was run with the same uncertainties as for the ground-based inversion, and which showed very little change in the posterior versus the prior fluxes, the prior uncertainty was increased to 100%. The prior error covariance matrix, **B**, was calculated using the square of the prior uncertainties in each grid cell as the variances and the co-variances were calculated assuming that the correlation between two grid cells decays exponentially with a correlation scale length of 200 km, and the correlation between flux time steps decays exponentially with a correlation scale length of 28 days.

### 3.2 Results and discussion

### 3.2.1 Inversion diagnostics

The TROPOMI inversion was run for 25 iterations and the ground-based observation inversion was run for 30 iterations. For the TROPOMI inversions, 25 iterations was deemed a sufficient number for convergence based on the change in the cost at each iteration, which was <1% after 17 iterations. For the ground-based observation inversion, the change in cost was only consistently <1% after 23 iterations (the cost at each iteration for both inversions is shown in Supplementary Fig. S3).

Another inversion diagnostic that is often used to determine the appropriateness of the state space and observation space uncertainties is the reduced chi-squared value, which is twice the final cost (see Eq. 14) divided by the number of degrees of freedom, which has an expected value of one (Tarantola, 2005). However, the reduced-chi-square criterion can be ambiguous as pointed out by Chevallier et al. (2007). In any case, we report here the reduced chi-square values for the TROPOMI and ground-based inversions, which were 1.08 and 2.16, respectively.

### 300 3.2.2 Modelled XCH$_4$

Column mixing ratios of CH$_4$ were modelled for each of the super-observations using the set-up described above. The observations for all months show high XCH$_4$ values for the southern part of the domain, especially in northern China. This is also captured, but with lesser magnitude, in the prior and posterior modelled XCH$_4$ (Fig. 2). In the summer months (June to August) there is also elevated XCH$_4$ in the central part of the domain, corresponding to the location of wetlands but also to oil

and gas fields. The posterior modelled XCH$_4$ had a much closer agreement with the observations, as expected. For example, for March the a posteriori Mean Error (ME) and Root Mean Square Error (RMSE) was 2 and 16 ppb, compared to that a priori with 45 and 50 ppb, respectively. For July, the a posteriori ME and RMSE was 3 and 13 ppb, compared to that a priori with 12 and 20 ppb. The differences between the posterior and prior modelled XCH$_4$ are shown in Supplementary Fig. S4.

The generally too high modelled XCH$_4$ using the prior state vector, especially in March, was primarily due to a too high background estimate when this was based on initial mixing ratios from EGG4. Further simulations using the CAMS greenhouse gas inversion product (CAMSv20r1) showed that the modelled XCH$_4$ is strongly sensitive to the fields of initial mixing ratio used, and using CAMSv20r1 the prior modelled XCH$_4$ was considerably lower (see Supplementary Fig. S5). The reason is the different vertical distributions of CH$_4$ in CAMSv20r1 versus EGG4 (see Supplementary Fig. S6), which convolved with the

averaging kernel and the FLEXPART-calculated averaging matrix, $\mathbf{H}^{ini,col}$ lead to quite different values for the background column average mixing ratio. A bias is also seen when XCH$_4$ is calculated directly from the initial mixing ratio fields (by applying Eq. 2) with EGG4 resulting in significantly higher, and CAMSv20r1 resulting in significantly lower, XCH$_4$ compared to the observations in March, but with smaller biases in July (see Supplementary Fig. S7). For this reason, the boundary conditions (i.e., 3D fields of initial mixing ratios) are optimized in the inversion simultaneously with the fluxes.


In the inversion using EGG4, the posterior scalars of initial mixing ratios were decreased in the latitude band 30°-90°N at all altitude layers and timesteps, while the mixing ratios in the band 0°-30°N were increased slightly in the lowest altitude layer and more strongly in the upper two layers, while the scalars for the Southern Hemisphere did not differ significantly from the prior value of 1.0 (Fig. 3). In contrast, in the inversion using CAMSv20r1, the scalars for 30-90°N were decreased only for the

lowest altitude layer, remained close to the prior value for the mid layer and increased for the uppermost layer (see Supplementary Fig. S8). Despite the very different background estimates using EGG4 versus CAMSv20r1, the inversions resulted in very similar posterior fluxes, which indicates that the optimization of the boundary conditions is successful in minimizing biases due to these (see Supplementary Fig. S9).

Figure 4 shows the area-weighted mean $XCH_4$ for the domain for 2-weekly intervals from March to October. The prior modelled $XCH_4$ follows the prior background estimate, which is driven by variations in the boundary conditions (based on EGG4) and differs considerably from the variation in the observed $XCH_4$. After optimization, the modelled $XCH_4$ more closely follows the observations, which is largely due to the improvement to the background estimate. Both the prior and posterior modelled $XCH_4$ remain close to their respective backgrounds until late April, when fluxes in the domain start to increase and

thus have a more significant impact on $XCH_4$ and return towards their background estimate after September.

### 3.2.3 Posterior fluxes and uncertainty reduction using TROPOMI

Figure 5 shows the mean posterior fluxes estimated from the inversion using TROPOMI observations as well as the posterior minus prior differences (flux increments). Overall, the posterior fluxes remain very close to those of the prior and the total mean posterior source over the domain for March to October is $30.3 \pm 22.7$ Tg/y compared to the prior estimate of $31.0 \pm 24.5$

Tg/y. The seasonal cycle also remained close to the prior estimate with a maximum in late July to early August (Fig. 6). The inversion did, however, reduce emissions for a few hotspots in northwestern Siberia in grid cells with important oil and gas sources, and increase emissions for a few hotspots in northern China, again in grid cells with important oil and gas sources.

It must be noted, however, that the uncertainty reduction on the fluxes (calculated as one minus the ratio of the posterior to

prior flux uncertainty) is quite small and mainly limited to the areas of the Western Siberian Lowlands and to the southern part of the domain, where it reaches 20-50% (Fig. 7). This is similar to the results of Tsuruta et al. (2023), who likewise found limited uncertainty reduction for the high northern latitudes using TROPOMI and little difference between the prior and posterior fluxes for their region of Eurasia (which included Fennoscandia). This is partly due to the poor observational coverage over Siberia where even outside of the winter season the number of observations is still limited (especially >50°N) and can be

due to frequent cloud cover (Gao et al., 2023). (The low uncertainty reduction is discussed further in Section 3.2.4). The pattern of uncertainty reduction in our study is persistent for all months and is largely determined by the distribution of observations and of the prior flux uncertainty. The more southern part of the domain is better covered by observations, especially over

Kazakhstan and northern China, while the prior flux uncertainties followed the distribution of the prior fluxes with larger uncertainties in the area of the Western Siberian Lowlands and for grid cells with hotspot emissions (see Supplementary Fig. S10).

### 3.2.4 Comparison with ground-based data inversions

Figures 5d and 5e show the posterior fluxes and flux increments, respectively, from the inversion with ground-based observations. The mean posterior fluxes show large emissions over the Western Siberian Lowlands, and generally higher emissions than in the prior estimate. They also indicate that some of the hotspot emission sources in northwestern Siberia are too large in the prior, which is consistent with the result of the inversion using TROPOMI. Moreover, the posterior fluxes indicate larger emissions for a few hotspots in southern Siberia coinciding with grid cells where there are coal mines. The mean posterior source over the domain from March to October is $34.6 \pm 10.2$ Tg y$^{-1}$. The difference between the posterior fluxes from the inversion using ground-based observations versus that using TROPOMI (Fig. 5f) follows a very similar pattern to the posterior minus prior flux increments (Fig. 5d), as expected, since the posterior fluxes from the inversion using TROPOMI are very close to the prior.

The ground-based observation inversion indicates an earlier and more intense summer maximum compared to the prior estimate and to the estimate from the TROPOMI based inversion (Fig. 6). In the ground-based inversion the maximum occurs in early July, versus late July to early August in the prior, and reaches a maximum of 51.8 Tg y$^{-1}$ for the mean of July versus 37.8 Tg y$^{-1}$ in the prior.

Moreover, the inversion using ground-based observations is better constrained than that using TROPOMI, and there are uncertainty reductions of up to 50% over a significant part of Western Siberia, corresponding to where the continuous measurement sites are located, although some gaps remain (Fig. 8). On the other hand, the southern and eastern parts of the domain are not well constrained.

As a further check on the inversion using TROPOMI observations, we compared the mixing ratios modelled using the prior fluxes, the posterior fluxes from the ground-based inversion, and the posterior fluxes from the TROPOMI inversion, against observations at all ground-based sites (Fig. 9). In this comparison, the optimized boundary conditions were used to show only the differences due to the fluxes. Overall, there is an improvement in the fit to the observations using the posterior fluxes from the ground-based inversion compared to the prior fluxes as would be expected since these observations were used in the inversion. The RMSE over all observations was reduced from 37 ppb a priori to 24 ppb a posteriori. However, using the posterior fluxes from the inversion using TROPOMI observations, did not lead to an improvement (nor a deterioration) in the fit to the observations, which is simply because the posterior fluxes in this case remained very close to the prior.

The reason for the lower uncertainty reduction (and smaller flux increments) using TROPOMI versus ground-based observations is essentially two-fold: i) The TROPOMI column average observations have larger uncertainties compared to the ground-based observations (in this study the median uncertainty used for ground-based observations was 12 ppb versus 16 ppb for TROPOMI) and the model-observation errors are weighted by the inverse of the square of the observation uncertainties (see Eq. 15). ii) Since satellite observations are of the total atmospheric column, the air masses in the columns can have more diverse source regions resulting in column SRRs that are more spread-out compared to those of point observations. This tends to lead to smaller deviations over background for the satellite observations compared to the point observations and, since the cost function depends on the square of the model-observation differences, a few large differences have more influence than a larger number of small differences. Thus, to compensate for the weaker constraint of each individual retrieval on the fluxes, many more retrievals are needed to achieve a similar constraint as provided by the point observations. Although this applies for any region, this is more notable in higher latitudes where there are generally fewer observations compared to mid-latitudes, and where the background uncertainty is larger meaning that even greater departures in the modelled mixing ratios from the background mixing ratio would be needed to constrain the fluxes.

## 4. Summary and conclusions

We have developed an efficient method to model total column observations, such as those from satellites, for Lagrangian Particle Dispersion Models (LPDMs) and, furthermore, to compute Jacobian matrices describing the relationship between fluxes and the change in the column average mixing ratio as needed in inverse modelling. This method means that the computations are in principle no more costly than those for point observations. The development will enable a more accurate representation of satellite observations (especially high-resolution ones) through using LPDMs and thus help to improve the accuracy of emission estimates obtained via atmospheric inversions.

Since LPDM backwards calculations are still needed for each observation, the computational cost is a limiting factor for using this method on the global scale for satellites providing a very large number of retrievals, e.g. TROPOMI which provides ~100,000 retrievals globally each day. This limitation can be overcome though by using "super observations", that is averages of retrievals, which reduces the number of calculations required. On the other hand, our method using an LPDM is well suited for regional inversion studies especially with observations from flux mapping satellites with a relatively high resolution, such as the TROPOMI XCH$_4$ product with a resolution of $5.5 \times 7$ km, MethaneSAT with a resolution of $0.1 \times 0.4$ km, the recently launched GOSAT-GW which has a resolution of $1 \times 1$ to $3 \times 3$ km resolution in Focus mode, and the future mission, CO2M with $2 \times 2$ km.

We presented a case study using the methodology to estimate CH$_4$ fluxes over Siberia using WFMD retrievals of XCH$_4$ from the TROPOMI instrument. We found that, for this northern region, the boundary conditions have a strong influence on the

modelled column mixing ratios, but by optimizing the boundary conditions any bias in these does not contribute to a bias in the posterior fluxes. Moreover, we compared the inversion with TROPOMI to one using ground-based observations. The ground-based observations provide a stronger constraint on the fluxes and greater uncertainty reduction compared to TROPOMI for this northern region. Although the posterior fluxes obtained using TROPOMI remained close to the prior, there were some consistent results with those obtained using ground-based observations, namely, a decrease in hotspot emissions in northern Siberia and an increase in a hotspot emission in northern China compared to the prior emissions.

Based on these results, the caveats of using satellite retrievals in regional inversions at high latitudes are: 1) the strong dependence of the modelled column mixing ratios on the boundary conditions and hence the need to set large uncertainties for the optimization of the boundary conditions, which has the effect of reducing the constraint of the observed column average mixing ratios on the fluxes, and 2) the limited observational constraint of the column average mixing ratios on surface fluxes in Siberia and hence low uncertainty reduction from inversions.

**Author contribution**

RLT designed the algorithms, wrote the code, ran the ground-based observation inversions and wrote the manuscript. NK ran FLEXPART for the satellite retrievals, ran the satellite-based inversions, and contributed to the manuscript. PS, IP, KS, SP, AS and MS provided advice on the use of the satellite data and modelling with FLEXPART and contributed to the manuscript.

**Competing interest statement**

The authors declare that they have no conflict of interest.

**Data availability**

TROPOMI WFMD retrieval data as well as the corresponding data documentation are available from the University of Bremen at https://www.iup.uni-bremen.de/carbon_ghg/products/tropomi_wfmd/. The JR-STATION data are available from the Global Environmental Database, hosted by ESD, NIES: http://db.cger.nies.go.jp/portal/geds/index. The other ground-based observations are available from the World Data Centre for Greenhouse Gases (WDCGG): https://gaw.kishou.go.jp. The FLEXINVERT and FLEXPART codes used for this study are available from the GitLab repository: https://git.nilu.no/flexpart.

**Acknowledgements**

This study was supported by the ReGAME project funded by the Research Council of Norway (grant no. 325610) and by the Horizon Europe project, EYE-CLIMA (grant no. 101081395). AS was partly supported by the Austrian Research Promotion

Agency under the project GHG-KIT (ID 42635422). We would like to acknowledge Oliver Schneising (University of Bremen) for providing the TROPOMI WFMD retrievals. Development of the TROPOMI WFMD product was supported by the European Space Agency via the projects GHG-CCI+, MethaneCAMP, and SMART-CH4 (ESA contract nos. 4000126450/19/I-NB, 4000137895/22/I-AG, and 4000142730/23/I-NS) and the Bundesministerium für Bildung und Forschung within its project ITMS (grant no. 01 LK2103A).

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

**Table 1: Ground-based atmospheric observation sites. The altitude refers to that of the air intake.**

| Station ID | Station name, country | Measurement type | Latitude (°N) | Longitude (°E) | Altitude (masl) | Network/Institute |
|---|---|---|---|---|---|---|
| UUM | Ulaan Uul, Mongolia | Flask | 44.45 | 111.09 | 1007 | NOAA |
| TER | Teriberka, Russia | Flask | 69.20 | 35.10 | 40 | Voeikov Main Geophysical Observatory |
| CPA | Cholpon-Ata, Kyrgyzstan | In-situ | 42.64 | 77.07 | 1613 | Agency on Hydrometeorology under Ministry of Emergency Situations of the Kyrgyz Republic |
| AZV | Azovo, Russia | In-situ | 54.71 | 73.03 | 110 | JRSTATION |
| BRZ | Berezorechka, Russia | In-situ | 56.15 | 84.33 | 168 | JRSTATION |
| DEM | Demyanskoe, Russia | In-situ | 59.79 | 70.87 | 63 | JRSTATION |
| KRS | Karasevoe, Russia | In-situ | 58.25 | 82.42 | 76 | JRSTATION |
| NOY | Noyabrsk, Russia | In-situ | 63.43 | 75.78 | 108 | JRSTATION |
| VGN | Vaganovo, Russia | In-situ | 54.50 | 62.32 | 192 | JRSTATION |

**Table 2: Overview of flux estimates used in the prior**

| Source type | Description | Resolution | Total for domain (Tg CH$_4$) |
|---|---|---|---|
| Anthropogenic | EDGAR-v8 | 0.1°, annual | 12.8 |
| Peatlands, wet and inundated soils, soil sink | LPX-Bern | 0.5°, monthly | Peatlands: 11.5 Wet and inundated soils: 5.8 Soil sink: -2.8 |
| Biomass burning | GFED-v4.1s | 0.25°, monthly | 1.2 |
| Ocean | Weber et al. (2019) | 0.25°, monthly | 0.5 |
| Geological | Etiope et al. (2019) | 1.0°, annual | 1.6 |

**Figure 1: Map of the inversion domain indicating oil & gas extraction and coal mining locations and the ground-based sites used in the inversion. The oil & gas and coal mining data were obtained from Global Energy Monitor (https://globalenergymonitor.org).**

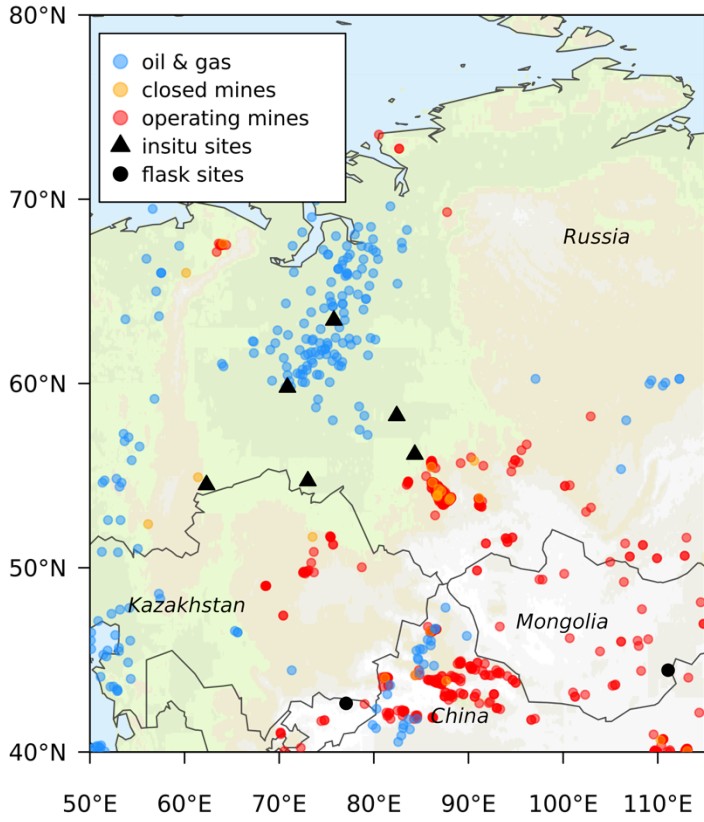


**Figure 2: Monthly mean XCH₄ from super-observations in ppb for March 2020 from a) observations, b) modelled using prior fluxes, and c) modelled using posterior fluxes and scalars of initial mixing ratios. Panels d) to f) are as for a) to c) but for the monthly mean XCH₄ for July 2020. Note that for plotting, the super-observations were averaged to a regular grid of 0.25°.**

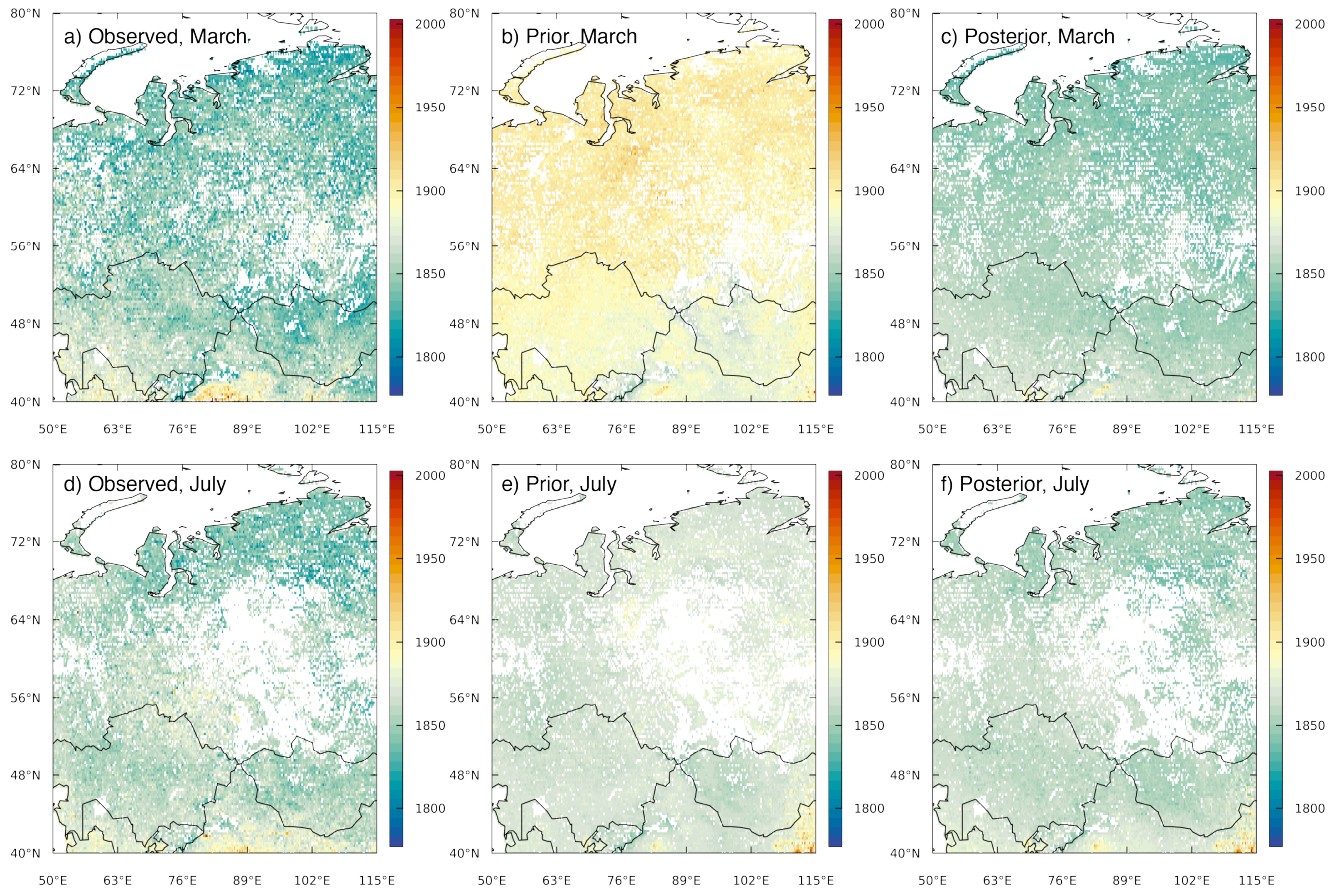


**Figure 3: Posterior scalars of the initial mixing ratios from the TROPOMI inversion using 3D initial mixing ratio fields from EGG4. In each sub-panel, the scalars are shown for each timestep of 28 days and for each of the four latitude bands.**

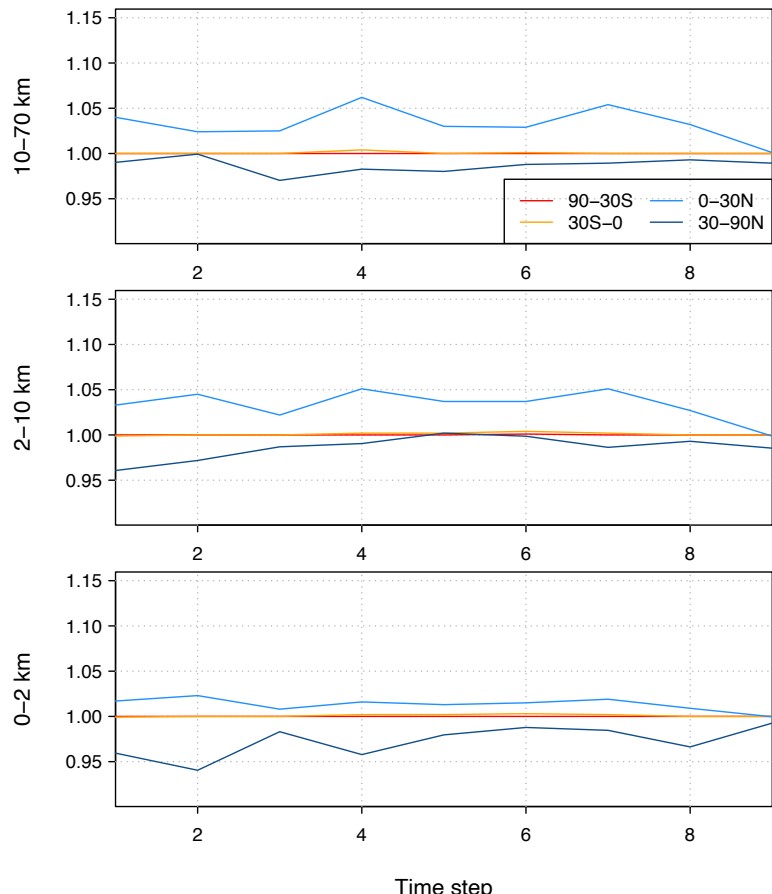


**Figure 4: Area-weighted mean XCH₄ for two-weekly intervals integrated over the domain for the inversion using EGG4 for the boundary conditions. The shading shows the area-weighted standard deviation of XCH₄ in each two-weekly interval over the domain.**

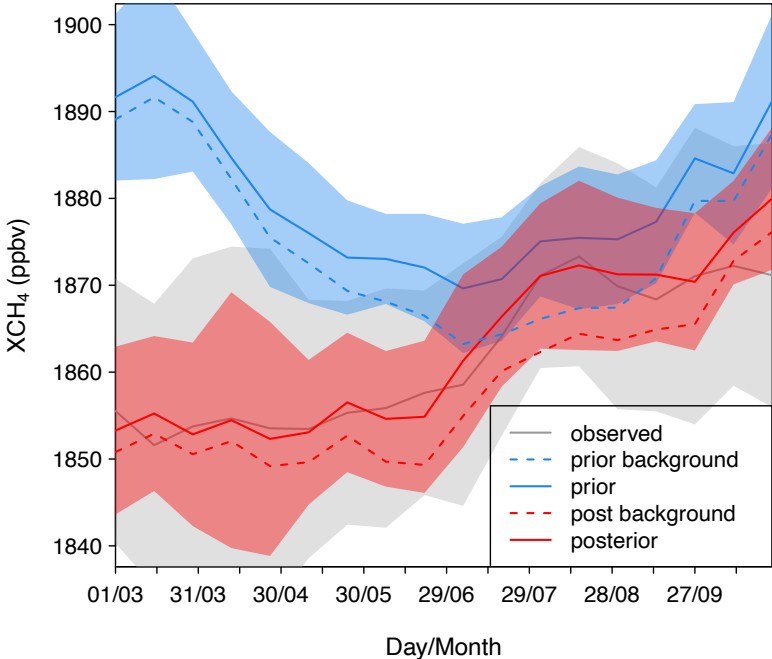

**Figure 5: Mean CH₄ flux and flux increments (units of g/m²/day) from the inversions. a) prior fluxes, b) posterior flux using TROPOMI,  c) posterior-prior flux increments using TROPOMI, d) posterior flux using ground-based observations, e) posterior-prior flux increments using ground-based observations and f) the difference in posterior fluxes using ground-based observations versus TROPOMI.**

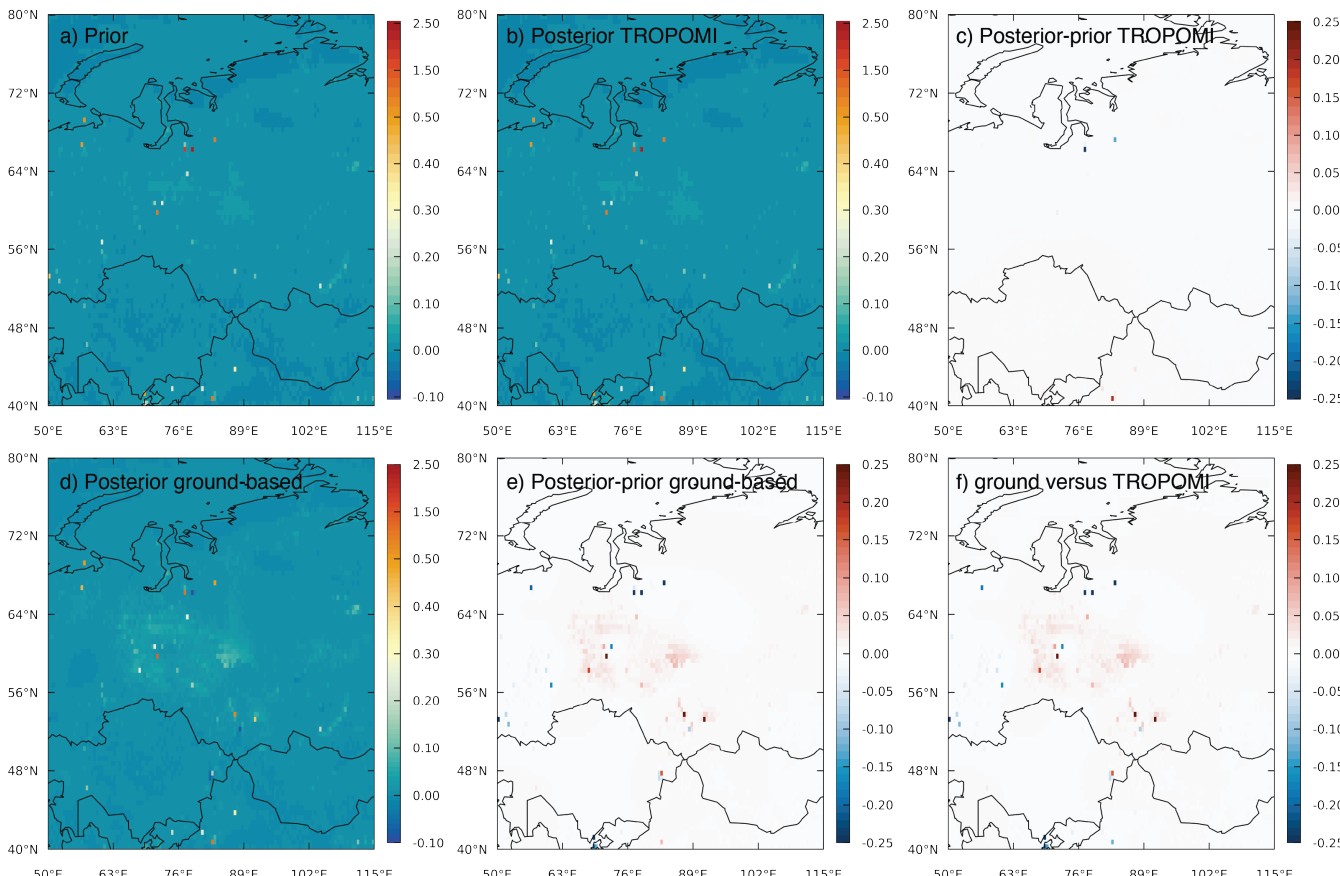


**Figure 6: Total CH₄ source for the domain shown 2-weekly for the prior estimate and for the posterior estimates from the TROPOMI and ground-based observation inversions.**

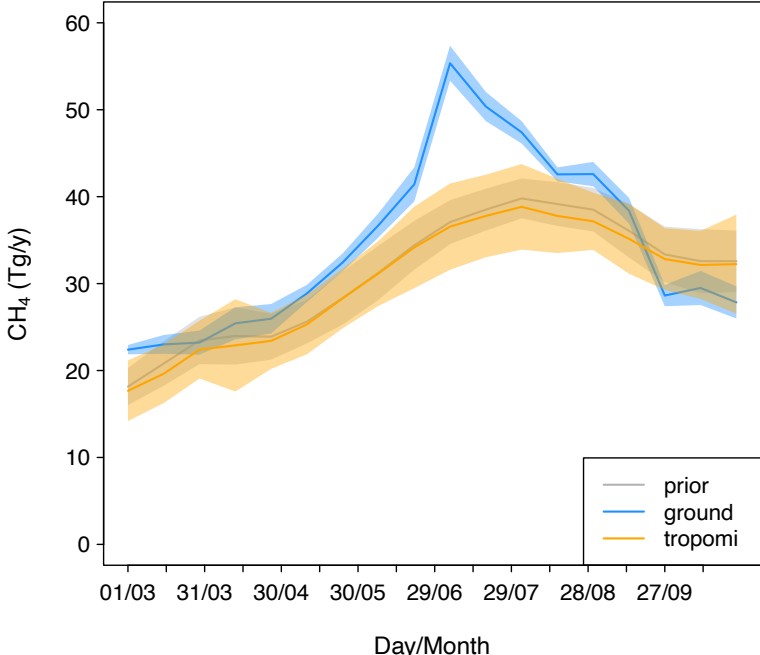

**Figure 7: Uncertainty reduction for the inversion using TROPOMI a) mean of March to October, b) mean March to May, c) mean June to August.**

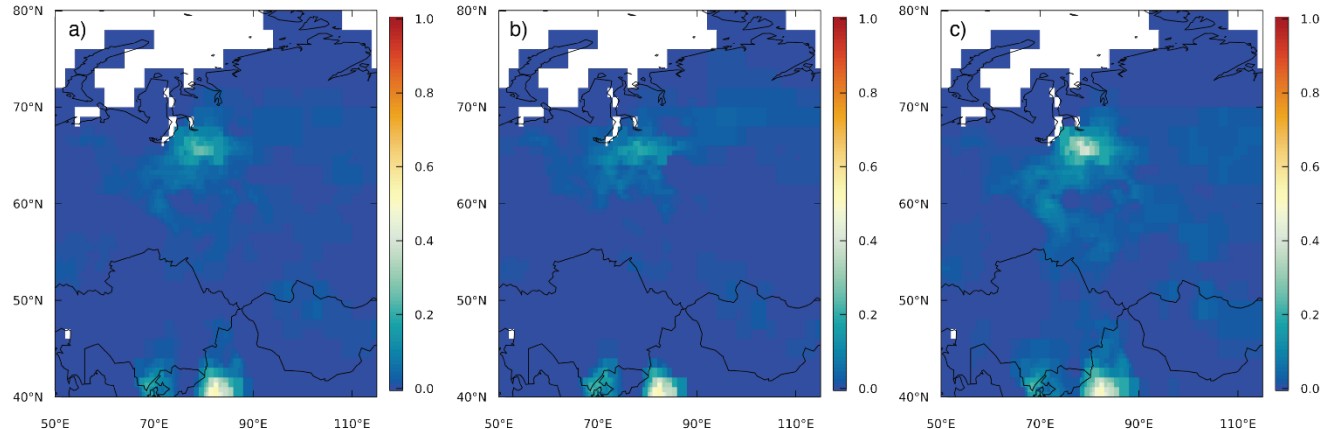

**Figure 8: Uncertainty reduction for the inversion using ground-based observations. The observation sites are indicated by the yellow circles.**

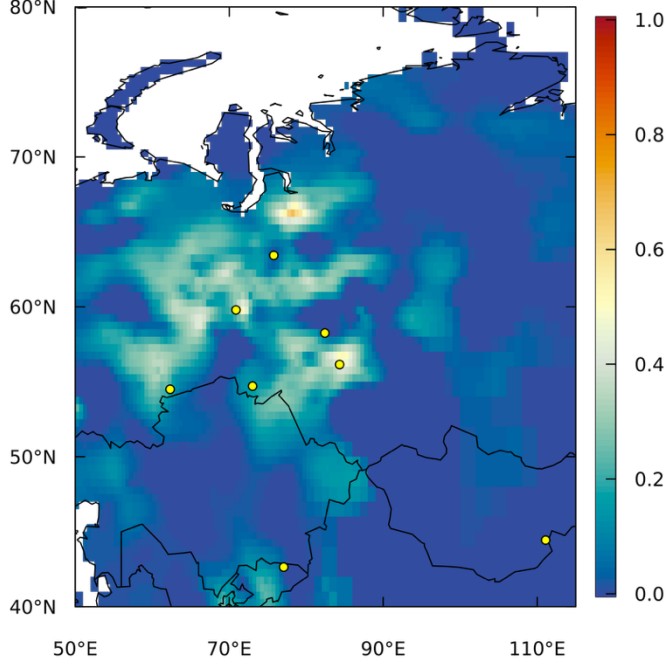

**Figure 9: Taylor diagram for the comparison of modelled versus observed CH$_4$ mixing ratios at ground-based sites. The angle gives the Pearson's correlation and the x-axis gives the normalized standard deviation for the comparison. Each point represents a site and the colour of the point indicates the modelled data used (prior: using the prior fluxes, ground: using the posterior fluxes from the inversion with ground-based observations, tropomi: using posterior fluxes from the inversion with TROPOMI observations). All model simulations used optimized boundary conditions to compare differences solely due to the fluxes used.**


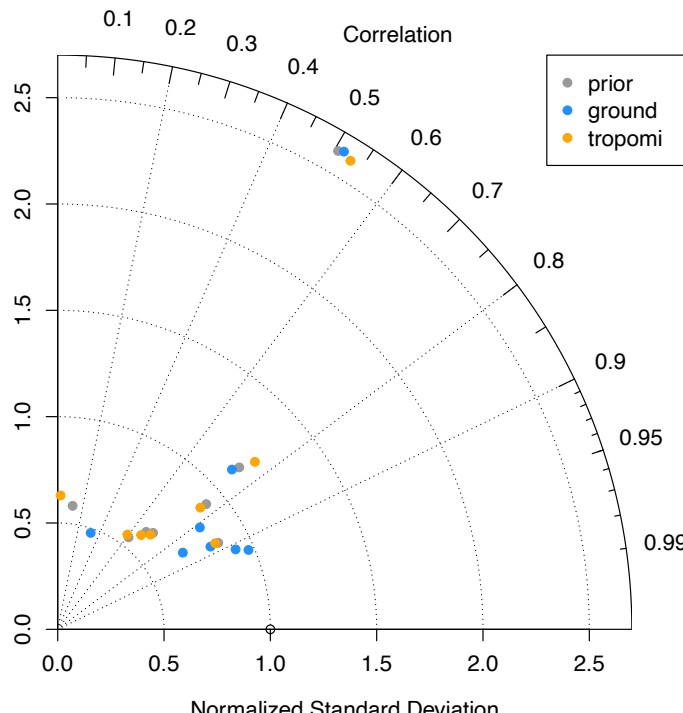
