# Peer review of "Efficient use of a Lagrangian Particle Dispersion Model for atmospheric inversions using satellite observations of column mixing ratios"

_EGUsphere, 2025_

## Referee Comment (RC3)

**Referee report:**

**'Efficient use of a Lagrangian Particle Dispersion Model for atmospheric inversions using satellite observations of column mixing ratios'**

**Author(s): Rona Louise Thompson et al. (2025)**
**MS No: egusphere-2025-147**
**MS type: Research article**

The paper describes further developments in the FLEXPART 10.4 code for simulating total column $CH_4$ mixing ratios. This implementation is important, as the use of satellite information is becoming increasingly valuable for identifying $CH_4$ hotspots and deriving emissions through inverse modelling. While the Siberia case study is interesting, and the integration of TROPOMI satellite measurements into FLEXPART is a valuable contribution, the methodology described in the manuscript is vague. It is too brief, the equations are confusing and poorly described, and there are several errors. Given that the authors claim this is the first study to calculate source-receptor relationships (SRRs) for column-averaged mixing ratios observed by satellites, a more thorough explanation of the methodology is needed. I believe the paper has potential; however, in its current form, I do not recommend it for publication unless the authors address all major comments listed below.

**Major revisions:**

1. As soon as I started reviewing this article, my first question was how the changes in this version of FLEXPART 10.4 differ from the recent release of FLEXPART 11. Since some of the co-authors of this paper also contributed to a recent publication - Bakels et al. (2024), which also demonstrates how FLEXPART can be used to simulate total column source-receptor relationships, it is unclear why the authors do not reference that work. For readers and new users of FLEXPART, it would be helpful if the authors explained the key differences between the two versions. Which version of the code should be used for simulating total column source-receptor relationships for satellite data?

2. In Section 2.1, the authors attempt to explain how total column observations were modelled using FLEXPART, but they do not specify which subroutines were modified. For clarity and reproducibility, it would be helpful to identify the main changes in the code. For example, mentioning specific subroutines - such as readreleases.f90, which handles the RELEASES input file - would be valuable for readers and new users. A brief description of the implemented changes, including key equations and code modifications, would enhance the utility of this work. It is possible that some of these details were presented in Bakels et al. (2024), but no reference is provided.

3. One of the main concerns is how some equations are presented in the article. For example, the authors state that the column SRR in FLEXPART 10.4 can be obtained using Equation (6).

$$H_{ijk}^{col} \sum_{n=1}^{N} a_n w_n \frac{t}{m_n \rho_{ijk}} \sum_{p=1}^{J_{i,jk,n}} m_p$$

Here, the authors describe $t$ as the sampling time, but my understanding is that $t$ in the equation should be interpreted as $\Delta t_{p,i,j,z\leq h}$. In other words, the sensitivity of the receptor located at $x_r$ at time $t_r$ to surface fluxes originating from $x_i$, $y_j$ is obtained by summing the time spent by particle $p$ over the grid cell $i,j$ within the surface layer of height $h$ at each discrete time step (see Equation 4 in Wu et al., 2018), who implemented total column simulations using a different but similar LPDM model. This is also described in Seibert and Frank (2004), Equation 8, where $\Delta t'_{ijn}$ is defined as the residence time of a trajectory in the spatio-temporal grid cell (i,n).

4. Assuming that Equation 8 is correct, shouldn't equation 9 be expressed as follow:

$$H_{ijk}^{col} = \frac{t}{m_n \rho_{ijk}} \sum_{n=1}^{N} \sum_{p=1}^{J_{i,j,k,n}} \frac{P_n}{P} m_p$$

It is not clear how Equation 8 can be further simplified to:

$$H_{ijk}^{col} = \frac{t}{m_n \rho_{ijk}} \sum_{p=1}^{J_{i,j,k,n}} m_p$$

Equation 9 requires further explanation. It is unclear how the authors transition from Equation 8 to Equation 9. They mention that Equation 9 can be seen as analogous to the calculation for point observations, but how exactly?

Later is mentioned that total column of BRR (equation 10) for the grid cell ijk can be written as:

$$H_{ijk}^{col,\,ini} = \frac{1}{m} \sum_{n=1}^{N} \sum_{p=1}^{J_{i,j,k,n}} \frac{P_n}{P} m_p$$

I would have expected Equation 10 to be similar in form to Equation 8.

$$H_{ijk}^{col,\,ini} = \frac{t}{m_n \rho_{ijk}} \sum_{n=1}^{N} \sum_{p=1}^{J_{i,j,k,n}} \frac{P_n}{P} m_p$$

It is not clear how the final particle position is calculated in Equation 10.

5. The paragraph from lines 130 to 135 requires further clarification. I do not fully understand the following part: "*In FLEXPART, an observation can be represented by releasing virtual particles from a volume in which the particles are distributed randomly. However, the default is that this volume is aligned with the meridians and parallels. Therefore, we have implemented an affine transformation on the particle positions so that the volume they represent matches the geometry of the retrieval.*" This description is vague and confusing. It would be helpful if the authors

provided the explicit form (i.e., equations) of the affine transformation applied to the particle coordinates.

6. In section 2.2 (averaging of retrievals), the authors do not mention whether the super observations were created in a rectangular grid cell or not, or use the distorted rectangles given by the satellite grid-cells.

Equation 12 from Section 2.2 (Averaging of retrievals) needs further review, the third term of the equation is wrong. Shouldn't be:

$$- \frac{1}{S} \sum_{m=1}^{M} \sum_{n=1}^{N} s_m a_{mn} w_{mn} x_{mn}^{pri}$$

Why the term of equation 13, cannot be further simplified as:

$$- \frac{1}{S} \sum_{n=1}^{N} \overline{a_n}\, \overline{w_m}\, \overline{x_n^{pri}}$$

7. In Section 3.1.3. it is not clear if inversion uses temporal correlation. If not, why?

8. At the beginning of the results section, and before presenting the modelled posterior $XCH_4$ (Section 3.2.1), it would be helpful to include a subsection showing the convergence diagnostics of the inversion, particularly given the use of TROPOMI satellite data. I recommend that the authors report the evolution of the cost function during the optimisation, along with the final cost function value normalised by the number of observations (i.e., final cost function J/n), which should ideally approach 0.5 under Gaussian error assumptions. Including these diagnostics would help assess the quality of the inversion and whether the assumed error statistics are appropriate.

9. The authors mention in Section 3.2.2 that the posterior fluxes remain very close to the prior, with a total mean posterior source of 30.3 Tg yr$^{-1}$ compared to the prior estimate of 31.0 Tg yr$^{-1}$. This limited adjustment suggests that the observational constraint had minimal impact on the flux optimisation. I recommend that the authors provide an analysis of the inversion's sensitivity to prior and observation uncertainties. In particular, it would be helpful to include convergence diagnostics, as stated previously (e.g., cost function components and their evolution), to assess whether the inversion system is appropriately weighted and whether the observations are effectively contributing to the flux estimates.

10. In Section 3.2.2, it is also mentioned that the uncertainty reduction in the fluxes is quite small, indicating that the pattern of uncertainty reduction follows the distribution of observations and the prior flux uncertainty. However, if the prior uncertainties (i.e., variances in B) are too small, the inversion will not adjust the fluxes significantly, even if the observations suggest otherwise. Similarly, if the observation uncertainties (i.e., variances in R) are too large, the inversion will

down-weight the observational constraints, resulting in minimal deviation from the prior.

11. In Section 3.2.3, the authors state: "The difference between the posterior fluxes from the inversion using ground-based observations versus that using TROPOMI (Fig. 8c) follows a very similar pattern to the posterior-minus-prior flux increments (Fig. 8b), as expected, since the posterior fluxes from the inversion using TROPOMI are very close to the prior." This raises the question: are we truly learning anything from TROPOMI? As noted earlier, an analysis of the convergence diagnostics would provide greater confidence in the results. It remains unclear whether the assumed uncertainties are appropriate and whether they lead to reliable posterior flux estimates.

**Minor revisions:**

Please add a title to each panel in Figure 2 to clarify which maps correspond to the observations, prior, and posterior. The legend should also include units.

In Figure 1, the country names should be labelled for better reference - the text mentions China, but it is unclear which boundary the authors are referring to.

**Reference:**

Bakels, L., Tatsii, D., Tipka, A., Thompson, R., Dütsch, M., Blaschek, M., Seibert, P., Baier, K., Bucci, S., Cassiani, M., Eckhardt, S., Groot Zwaaftink, C., Henne, S., Kaufmann, P., Lechner, V., Maurer, C., Mulder, M. D., Pisso, I., Plach, A., Subramanian, R., Vojta, M., and Stohl, A.: FLEXPART version 11: improved accuracy, efficiency, and flexibility, Geosci. Model Dev., 17, 7595–7627, https://doi.org/10.5194/gmd-17-7595-2024, 2024.

Seibert, P. and Frank, A.: Source-receptor matrix calculation with a Lagrangian particle dispersion model in backward mode, Atmos. Chem. Phys., 4, 51–63, https://doi.org/10.5194/acp-4-51-2004, 2004.

Wu, D., Lin, J. C., Fasoli, B., Oda, T., Ye, X., Lauvaux, T., Yang, E. G., and Kort, E. A.: A Lagrangian approach towards extracting signals of urban CO2 emissions from satellite observations of atmospheric column CO2 (XCO2): X-Stochastic Time-Inverted Lagrangian Transport model ("X-STILT v1"), Geosci. Model Dev., 11, 4843–4871, https://doi.org/10.5194/gmd-11-4843-2018, 2018.

---

## Author Comment (AC1)

**Reply to review 2**

This work documents an interesting case study in which a Lagrangian Particle Dispersion Model (LPDM) is used together with high-resolution satellite data in an inversion. The paper describes an efficient method for producing SRR 'footprints' using an LPDM from a large number of satellite retrievals, which would ordinarily be resource-intensive and time-consuming. The new method is applied to a northern high-latitude region and results are compared to an inversion using ground-based observations within the same region. The authors found that the satellite-based inversion produced posterior fluxes which were very close to the prior, and put forward suggestions for why this was the case.

Overall this is an interesting study and a nice documentation of a method that will likely become more and more important as satellite resolution continues to increase. The method is well-described on the whole and the figures are OK. I have some comments and suggestions regarding some aspects of the method and the presentation, but I am happy to recommend publication of the manuscript after these are addressed.

We thank the reviewer for this comprehensive review. We insert our replies (indicated in blue) to each comment below.

**Major comment**

Throughout the document, the authors correctly refer to the fact that LPDMs are not usually used for this type of satellite-based inversion – e.g. 'LPDMs have not been used to any significant extent with satellite observations' (line 48).

However, I do think some discussion of any previous satellite-based LDPM inversions is merited. It should be made clear exactly what exactly the novel parts of this new methodology are. For example, Ganesan et al. (2017) performed a GOSAT-based inversion over India using a different LDPM. There should be some discussion of how your method differs to and updates theirs, as is necessary due to the increased volume of data available from TROPOMI compared to GOSAT. The novel aspects of your method should be highlighted more explicitly (e.g. the use of $P_n = Pa_n w_n$, in Eqn. 7, which is nice).

We agree that we should specifically mention the previous studies using Lagrangian models to model satellite observations and discuss how our study differs from these. We now refer to these studies in the introduction L51 and include a line after Eq. 5 discussing how our method compares to what has previously been done.

**Specific comments**

Line 85: 'SRRs have only been calculated for point observations …'. For FLEXPART, although this is not true of other Lagrangian models.

We were referring to FLEXPART, but we agree that this statement is misleading and that there are examples of studies using LPDMs to model satellite observations (e.g. studies

by Wu et al. 2018 and Ganesan et al. 2017). We have now corrected this statement to state emphasising instead the use of an LPDM with large numbers of satellite observations.

Line 101: How exactly might chemical loss be represented during the backward simulation? Via an assumed fixed lifetime, or via actual representation of OH concentration and chemical loss rates? Is there likely any effect from not including this here?

Linear chemistry can be accounted for in the backwards simulations. In this study, we have accounted for the loss of CH4 due to reaction with OH by using pre-calculated 3D and monthly fields of OH concentrations, which were computed by the GEOSChem model. The loss of CH4 due to OH in this study, however, is very minimal since the particle trajectories were only calculated for 20 days backwards in time. The effect of accounting versus not accounting for OH in this case is likely negligible owing to the high latitude of the domain and the length of the trajectory calculations.

Line 195: Is it possible to include a map of the variable resolution of the optimised flux grid?

We now include maps of the variable grid used for the TROPOMI and the ground-based observation inversions in the Supplement (Fig. S1).

Line 255: Can you provide a measure of the chi-square value for the various retrievals to test the convergence?

For the retrieval product that we use, namely WFMD v1.8, there is no chi-square value provided. Instead the data provider gives a binary quality flag (0 = good, 1 = bad) based on a number of criteria (Schneising et al. Atmos. Meas. Tech., 2019) and we have selected only the "good" data.

Line 295: I am a little surprised at the limited posterior uncertainty reduction. You say that this is to be expected due to limited observational coverage due to cloud cover and other issues, but you say earlier that there are over 3500 super observations per day, with TROPOMI having good sensitivity down toward the surface. Is the satellite coverage quite heterogeneous? Perhaps a figure showing the super observation coverage density by the satellite over the region for the spring and summer periods would be useful (perhaps alongside the SRR map in Figure S6).

We have included figures showing the number of observations per 14-day time interval for the TROPOMI super-observations and for the ground-based observations (see Supplementary Fig. S2). For the TROPOMI observations, we show the number per time interval as well as latitudinal intervals of 5° to indicate how the number of observations also varies by latitude. The number of observations strongly decreases with latitude, especially in March and October, as expected. There are also fewer observations from the mid May to mid July in the latitudes north of 50°N owing to cloud cover.

Concerning the low uncertainty reduction in the inversion using TROPOMI, this has a number of reasons other than the number of available observations. First, the TROPOMI XCH4 observations are more uncertain compared to the ground-based observations. The median uncertainty for XCH4 was 16 ppb compared to 8 ppb for the ground-based observations, and the model-observation errors are weighted by the inverse square of the uncertainty. Second, the satellite SRRs are smeared out over larger regions, compared to the SRRs for ground-based observations, which are focussed over smaller regions. This leads to stronger deviations in the modelled mixing ratios relative to the background (i.e., if there are sources in the SRR region) for the ground-based observations compared to the satellite observations. Since the cost function (in the inversion) includes the quadratic difference between observation and modelled mixing ratio, a few large differences have more impact than a large number of small differences. This means that the ground-based observations (with more focused SRRs) have more impact in the inversion.

Line 295: If you were expecting such poor satellite coverage over a difficult-to-observe region, why did you choose this region for your case study?

We were actually surprised that there is almost no uncertainty reduction over the domain in the inversion using TROPOMI. We had chosen this study domain as it extends as far south as 40°N meaning that at least in the southern part of the domain there should be sufficient number of observations, and because this domain contains quite important CH4 sources from coal, oil and gas, as well as important wetland fluxes (namely, in the Western Siberian Lowlands). A further, non-scientifc reason for choosing this domain was because the project funding this work was focused on high-latitude CH4 sources, so we were obliged to look for a domain including high latitudes.

Figure 10: I think there are some data points falling outside of the boundary of the Taylor diagram (around NSD = 2.5; R = 0.5). These are easily missed by the reader and the Taylor diagram should be expanded to include them if possible. Also please include a label on the angular axis.

We have expanded the radius of the Taylor diagram to include the points that were "off-scale" in terms of NSD. We also include the label on the angle, i.e., "Correlation".

Supplementary Figure: I'd be interested to see a plot showing 'posterior – prior' XCH4 for the results shown in Figure 2. Is there any spatial variability in the change to XCH4 produced by the inversion in this case or is it simply a homogeneous addition/subtraction of background $CH_4$ within that latitude band?

We have now included a figure in the Supplement (Fig. S4) showing the differences between the prior and posterior XCH4 for March and for July. There is some spatial variability in the change a posteriori relative to a priori: for March the increment is more negative in the north versus the south, for July the increment is small negative in the north and east and slightly positive in the west and south.

---

## Author Comment (AC2)

**Reply to Review 3**

We thank the reviewer for the comprehensive review and reply to each of the comments below.

Comment 1:

The paper by Bakels et al. (2024) describes the new FLEXPART v11. We presume the reviewer is referring to section 9.2 of Bakels et al. (2024), which describes how in FLEXPART v11 a simulation can be started using a custom particle initialization, which in principle could be used to represent satellite retrievals. However, there is no code provided in FLEXPART v11 to generate such a custom particle initialization, and modelling satellite observations is just mentioned as a possible application. In fact, this possible application is mentioned as it is exactly the work described in this manuscript, that is, how to make particle releases for a satellite retrieval in order to calculate total column SRRs. The reason why this manuscript refers to FLEXPART v10.4, and not to v11, is because the developments started before FLEXPART v11 was ready, and because only the FLEXPART v10.4 code is able to be used directly to calculate total column SRRs. It is planned, however, to release code in the future to initialize particle releases for satellite retrievals for use in FLEXPART v11.

Bakels et al. (2024) also describes the Linear Chemistry Module (LCM), which was implemented by me (Rona Thompson). The LCM is only a forward in time simulation mode and does not provide source receptor relationships (SRRs), which are needed for the inversion and are described in this manuscript. In the LCM mode, the whole domain is filled with virtual particles, which are then sampled to represent observations, either in-situ observations or satellite retrievals.

Comment 2:

The main purpose of the manuscript is to describe the methodology for calculating total column SRRs that could be implemented in any Lagrangian particle dispersion model, and not specific to FLEXPART. We prefer not to include details of the new routines in FLEXPART v10.4 in the manuscript, however, we now include these in the supplement.

Notice specifically that, in FLEXPART v11, there were no code changes made that are specifically related to modelling satellite column SRRs. However, the new v11 feature allowing to define particle releases with complete flexibility (section 9.2 of Bakels et al., 2024) makes it possible to define these releases, exactly as described in this study, such that FLEXPART v11 will determine satellite column SRRs. One may therefore consider this study as describing the pre-processing step that is necessary to produce the FLEXPART v11 input files for satellite column SRRs. In FLEXPART v11 itself, no changes are needed, other than for the output of the SRR files for satellites.

Notice also that this could be used, for instance, to provide corresponding input data files to produce SRRs for slant column measurements. Again, no changes would be needed in FLEXPART v11 (other than for the output of the SRR files for satellites), and

the particle release files could be produced in a very similar way as described in this paper – just accounting for the additional complexity of the slant column geometry.

Comment 3:

In Eq. 6, $t$, is the residence time of the particles in the grid cell as also described in Seibert and Frank (2004). We have modified Eq. 6 and instead of mass we now use the transmission function, which represents the fraction of mass remining in a particle which may change if there is atmospheric chemistry (as described in Seibert and Frank (2004)). We have also changed the following equations for the calculation of $\mathbf{H}^{col}$ accordingly.

Comment 4:

We have updated the equations (as mentioned in response to the previous comment). The main point is that the equation for the total column SRR can be expressed as a sum over all particles in all layers when the information on how each layer's SRR ($\mathbf{H}_n$) contributes to the total column, i.e. the weighting by $a_n w_n$, is incorporated into the initialization of the particles. Here we incorporate that information by varying the particle density in each layer where the number of particles in layer $n$ is $P_n = P a_n w_n$ where $P$ is the total of number of particles released per retrieval.

In Eq. 11 (formerly Eq. 10) there is no division by density, since what is wanted is the number of particles terminating in a given grid cell relative to the total number of particles released. This gives a unitless fraction. This fraction is calculated for all grid cells in 3D space and together defines a weighting matrix for the influence of the initial mixing ratios on the column observation.

Comment 5:

In the original FLEXPART code, releases could only be made for rectangular volumes which were aligned with the meridians and parallels (the particles are randomly distributed in these volumes). However, the satellite pixel geometry can be rotated with respect to the meridians and parallels and may not be rectangular. In the case of TROPOMI, the pixels are trapezoids and are rotated. Therefore, to make a release that represents the exact geometry of the satellite pixel, we perform an affine transformation. This involves the following steps:
1. Calculate the angle of rotation of the satellite pixel relative to the meridians
2. Calculate the equivalent lat and lon bounds of the unrotated pixel
3. Calculate the angle of distortion from satellite pixel
4. Calculate the equivalent lat and lon bounds of the undistorted and unrotated pixel
5. Distribute the particles randomly within this rectangular volume
6. For each particle, reapply the rotation and distortion so that the particles all fall within the original satellite pixel.

This is performed in the subroutine releaseparticles_satellite.f90 which is available from the open gitlab repository: https://git.nilu.no/flexpart/flexpart.git

We now include the above information in the supplement. Since the affine transformation is a whole algorithm we do not include all the equations in the supplement but refer the reader to the code in the aforementioned repository.

Also notice again that all of this can be considered as a pre-processing step for defining the particle release file in FLEXPART v11, since in this version the individual particle release positions can be specified in a file part_ic.nc. These positions are completely independent of a particular grid geometry. This was not possible in the older FLEXPART v10.4 version still used in this paper.

Comment 6:

The super observations are created for rectangular grid cells, which are aligned with the meridians and parallels. We now specify this in Section 2.2.

Indeed, the denominator of the third term on the RHS of Equation 12 should be S and not M. This is a typo in the writing of the equation only, the expression in the code is the correct one.

Equation 14 (formerly Eq. 13) cannot be simplified further, as the reviewer suggests, because:

$$\frac{1}{S}\sum_{m=1}^{M}\sum_{n=1}^{N} s_m a_{mn} w_{mn} x_{mn}^{pri} \neq \sum_{n=1}^{N} \overline{a_n w_n}\, \overline{x_n^{pri}}$$

since $x^{pri}_n$ and $a_n$ are unique for each retrieval. (Note division by S on the RHS is not needed since the mean of $a_n$, $w_n$ and $x^{pri}_n$ are used).

Comment 7:

The prior error covariance matrix also includes temporal correlation of prior flux errors. The temporal correlation is calculated using exponential decay with time with a correlation length of 28 days. We have added this information to Section 3.1.3.

Comment 8:

We include a short new section, 3.2.1, on the inversion diagnostics and put figures of the cost per iteration in the Supplement.

We think the reviewer is referring to the reduced chi-squared value, which is twice the final cost divided by the number of degrees of freedom, which has an expected value of one if the uncertainties are appropriately chosen (Rodgers, 2000; Tarantola, 2005). In practice, however, it is not always possible to achieve a value of one, and as pointed-out by Chevallier et al. (2007), the reduced-chi-square criterion can be ambiguous and alone is not a sufficient criterion for assessing the appropriateness of the uncertainties. In any case, the reduced chi-square values were as follows:

1) using TROPOMI: 1.08

2) using ground-based observations: 4.86

Comment 9:

Indeed, the TROPOMI observations provide little constraint on the fluxes at this high latitude. The reason for this is that the TROPOMI XCH4 observations at this latitude are more representative of background air and not so sensitive to fluxes at the surface. For spring and autumn, depending on atmospheric temperature structure, this can be because the boundary layer is shallow, and below the level of peak sensitivity for the TROPOMI instrument (as can be seen from the averaging kernels).

The observation uncertainties (i.e., for the super-observations) are determined from the retrieval information, namely, these are calculated as the area-weighted quadratic sum of the individual retrieval uncertainties (as determined from the retrieval algorithm). The median observation uncertainty was around 12 ppb and did not vary much between months. In addition, we consider a background uncertainty estimate. In total, the observation space uncertainties had a median value of 16 ppb and inter-quartile range of 14 – 18 ppb. We consider this to be a very reasonable estimate of the uncertainty of the column average mixing ratios at this latitude and not an overestimate. Of course, increasing the observation space uncertainty would further reduce the constraint on the fluxes.

We did perform a number of sensitivity tests but did not include these in the paper, since the focus is rather on the methodology of using satellite observations in an inversion based on a Lagrangian transport model. We tested using an uncertainty of 50% of the prior flux value (as was also used in the inversion using ground-based observations) and then because the posterior fluxes did not differ from the prior fluxes, we increased the uncertainty to 100%. The result remained the same, i.e., the posterior fluxes remained very close to the prior fluxes. We now include this information in Section 3.1.3.

Comment 10:

As we state in our reply to comment 9, we consider that the prior and observation uncertainties are appropriately chosen. Furthermore, for the inversion using TROPOMI we used twice as large prior uncertainties compared to the inversion using ground-based observations.

Comment 11:

At this northern latitude, the answer is that TROPOMI provides little constraint on the surface fluxes. This result has also been found by an independent study using Eulerian atmospheric transport model and an ensemble data assimilation algorithm (Tsuruta et al., Remote Sens., 15, 1620, https://doi.org/10.3390/rs15061620, 2023).

We have used our FLEXPART code and the inversion framework, FLEXINVERT, also in a study using TROPOMI over Europe and find there that TROPOMI provides a more

significant constraint on fluxes in southern, western and central Europe, but not a very large constraint in northern Europe.

However, in general it is to be expected that satellite observations have a weaker constraint on the fluxes than ground-based observations. First, the satellite observations are more uncertain compared to the ground-based observations (the median uncertainty for XCH4 in our study was 16 ppb compared to 8 ppb for the ground-based observations) and the model-observation errors are weighted by the inverse square of the uncertainty. Second, the satellite SRRs are smeared out over larger regions, compared to the SRRs for ground-based observations, which are focussed over smaller regions. This leads to stronger deviations in the modelled mixing ratios relative to the background (i.e., if there are sources in the SRR region) for the ground-based observations compared to the satellite observations. Since the cost function (in the inversion) includes the quadratic difference between observation and modelled mixing ratio, a few large differences have more impact than a large number of small differences. This means that the ground-based observations (with more focused SRRs) have more impact in the inversion.

Minor comments:

We have added titles to Fig. 2 and the units are given in the figure caption.

We have added country labels to Fig. 1

---

## Author Comment (AC3)

**Reply to reviewer 1**

We thank the reviewer for this comprehensive review. We insert our replies (indicated in blue) to each comment below.

**General comments**
This study present an innovative method for estimating total column methane ($XCH_4$) using a Lagrangian Particle Dispersion Model (LPDM), FLEXPART, and way to assimilate the data in an atmospheric inverse model, FLEXINVERT. The case study is carried out for Siberia for 2022. The comparison against the "traditional" ground-based inversion showed broad agreement with the inversion using TROPOMI data, and consequently reliability and a good potential in the presented method for estimation of regional $CH_4$ fluxes. The method sounds applicable for other LPDMs, and could contribute significantly to the atmospheric inverse modelling community with new ways to infer greenhouse gas flux information from satellite data. The fact that LPDMs can be run in much higher resolution than Eulerian transport models will be an advantage for incorporating information from future satellites with much higher spatial resolution. Therefore, this paper is worth of prompt publication after considering a few points below.

- The authors found large differences in modelled $XCH_4$ depending on background initial mixing ratios. The boundary conditions were optimised to somewhat discriminate the "errors", but how can it be sure that the signals within the domain is not over constrained by the background? In L280, it is said that the correction of modelled $XCH_4$ was "largely due to the improvement to the background estimate", and the posterior fluxes from the TROPOMI inversion did not change much from the prior. I suppose that with less uncertainty in the background, the fluxes would change more. How do you know what is a good balance?

Indeed, with more accurate background XCH4 estimates smaller uncertainties can be used for the scalars optimizing the initial mixing ratios, and thus the model-observation differences will have a stronger impact on the posterior fluxes.

The optimization of the initial mixing ratios is made by optimizing scalars, which are defined for boxes of the atmosphere (for a given number of latitudinal bands and vertical layers). Thus many retrievals in different parts of the domain will contribute to constraining the same scalars, because the BRR (background receptor relationship) is spread out over different latitudes and vertical layers. On the other hand, the SRR (source receptor relationship) is strongest in the vicinity of the retrieval. Thus there is a different constraint on the intial mixing ratios versus the fluxes. In this way, if accurate background estimates are obtained, even if large uncertainties are assigned to the scalars of initial mixing ratio these should not differ significantly from the prior value of one, and the model-observation differences will more strongly constrain the fluxes. In this way, it is not so important how large the uncertainties on the scalars of initial mixing ratio are, as long as they are sufficiently large to correct a wrong background. Please also see our reply to the comment on L209-213.

- High northern latitudes are challenging regions such that satellite retrievals are associated with various biases, especially those related to seasonal variations may plan an significant role. I wonder how much of the differences between the ground-based and TROPOMI inversions were due to these biases. Please discuss.

The posterior fluxes from the inversion using TROPOMI remained very close to the prior fluxes. This is simply because TROPOMI does not provide a strong constraint on surface fluxes at this latitude. Thus, the difference in posterior fluxes between the ground-based and TROPOMI inversions is simply due to the fact that in the TROPOMI inversion the fluxes do not significantly change with respect to the prior whereas in the ground-based inversions they do.

- The inverse model results are associated with various uncertainties. You have discussed and did sensitivity tests on background mixing ratios, but other optimization setups, such as choice of retrieval products, pre-processing methods of the satellite data, prior fluxes and prior uncertainties for observations and fluxes are also important. As this paper do not present variety of sensitivity tests, the conclusion of the paper about $CH_4$ flux estimates should be presented carefully that the results may change significantly depending on the setups.

We did in fact carry-out some sensitivity tests for the choice of prior uncertainties. We tested varying the prior uncertainties, and found the inversion results to be fairly insensitive to prior uncertainties chosen as 50% or 100% of the prior flux value.

For the observation space uncertainties for XCH4, these were determined for each super-observation as the quadratic sum of the individual retrieval uncertainties. This was a larger number than the standard deviation of individual retrievals going into each super-observation. The median observation uncertainty was around 12 ppb and did not vary much between months. In addition, we consider a background uncertainty estimate. In total, the observation space uncertainties had a median value of 16 ppb and inter-quartile range of 14 – 18 ppb. We consider this to be a reasonable estimate of the uncertainty of the column average mixing ratios. Increasing the observation uncertainties would not change the result of the TROPOMI inversion, since the posterior fluxes remain very close to the prior fluxes, and increasing the observation uncertainties would bring them even closer to the prior.

We also looked at the Official XCH4 retrieval product from SRON, however, we found that this contained many spurious XCH4 values over our domain, which was not the case for the WFMD retrievals. Furthermore, even the bias corrected XCH4 from the Official retrieval still contains biases due to surface albedo and aerosol size parameter (Balasus et al. 2024). Therefore, we chose to use the WFMD retrievals in our study.

Concerning the conclusions of our paper, we only state that the posterior fluxes remain close to the prior fluxes, and that there was limited uncertainty reduction. This result is

robust to the choice of prior uncertainty (50% or 100%) and would not change if larger observational uncertainties would be used, and we think smaller observational uncertainties would underestimate the uncertainty.

**Specific comments**

L160: Why do you use area-weighed averages? I understand it is somewhat reasonable for mixing ratios, but for averaging kernels and presser weighting, I do not fully understand how the area would be affected. Could you also explain how did you take into account differences in number of observations within the aggregated cells?

If one weights the mixing ratios by area, for consistency it is also necessary to apply this to the averaging kernels and pressure weighting.

The analysis column average mixing ratios are defined as:
$$\hat{x} = x_{pri} + A(x - x_{pri}) \tag{1}$$
The mean of column average mixing ratios is:
$$\bar{x} = \frac{1}{S}\sum_{m=1}^{M} \widehat{x_m} s_m \tag{2}$$
where $s_m$ is the area of the ground pixel of the retrieval. Substituting Eq. 1 into Eq. 2 and expanding gives:
$$\bar{x} = \frac{1}{S}\sum_{m=1}^{M} x_{pri,m} s_m + \frac{1}{S}\sum_{m=1}^{M} A_m x_m s_m - \frac{1}{S}\sum_{m=1}^{M} A_m x_{pri,m} s_m \tag{3}$$

The super-observations are calculated as the mean of all retrievals with the centre of the retrieval falling within the defined grid cell. The number of retrievals per super-observaition can of course can vary. We treat the errors of the individual retrievals as uncorrelated, namely the super-observation uncertainty is calculated as the quadratic sum of the individual retrieval uncertainties.

In the averaging algorithm, we also look at the distribution of retrievals within the grid cell as defined in the current iteration. If there are not retrievals in every quarter of the grid cell, then the grid cell is any how divided in the next iteration to avoid having a super-observation for a grid cell which is not fully represented by the retrievals. We have added this extra information to Section 2.2.

Section 3: Did you include temporal correlation of the state vectors? Please add information somewhere.

The prior error covariance matrix also includes temporal correlation of prior flux errors. The temporal correlation is calculated using exponential decay with time with a correlation scale length of 28 days. We have added this information to Section 3.1.3.

L195: Could you add a figure on spatial resolution? Where were lowest and highest resolutions? Were the resolutions same for the TROPOMI and ground-based inversions, despite the fact that they would have differences in "how strongly the fluxes influence the observations" due to differences in locations and quantity (surface vs total column) of the observations? Please clarify.

We have added figures showing the grid used for the state vector for the TROPOMI and for the ground-based observation inversions to the Supplement and refer to these in Section 3.1.1. The grid was calculated separately for the TROPOMI and for the ground-based observations, since these have different coverage and sensitivity to the fluxes.

L209-213: In later sections, it is said that the background mixing ratios were also optimised. What were the uncertainties in the boundary conditions?

For the TROPOMI inversions, we used a fairly large uncertainty for the scalars of the initial mixing ratios of 5%. This uncertainty was based on a number of tests, which showed that smaller uncertainties did not give sufficient freedom to adjust the background column mixing ratios, resulting in an erroneous seasonal cycle in the posterior fluxes with very large fluxes in spring and smaller fluxes in summer, which was owing to the large positive bias in the background mixing ratios in spring and small positive bias in summer.

For the ground-based observation inversions, the background mixing ratios were also optimized, but for these inversions the background estimate was more accurate and smaller uncertainties were used, of 1%.

We now include this information in Section 3.1.1.

L218: Why did you chose the grid cell sizes of 0.25° and 0.5°? FLEXPART is run at 0.5° and smallest optimisation spatial resolution is also 0.5°, so why did you chose to have observations at higher spatial resolutions? Can FLEXPART resolve differences well (or what is done) if there are more than one observations within a 0.5° x 0.5° grid cell?

In the Lagrangian model, there is no finite resolution or underlying grid for the modelling of the particle trajectories. Thus, the observations can be represented (via the release of virtual particles) as a point (as is done in the case of ground-based in-situ observations) or as a volume (as in dones for the satellite column observations). The 0.5° x 0.5° resolution refers to the grid resolution on which the particles are sampled, and thus the output resolution of the SRRs. By using a finer resolution for the particle releases the atmospheric transport at and around the actual observation can be better resolved.

L223: I suppose number of observation vary a lot within the study period. Please add information about number of observations also perhaps in 14-days temporal resolution, which is your flux optimisation resolution. I would also like to see for both TROPOMI and ground-based observations.

We have included figures showing the number of observations per 14-day time interval for the TROPOMI super-observations and for the ground-based observations (see Supplementary Fig. S2). For the TROPOMI observations, we show the number per time interval as well as latitudinal intervals of 5° to indicate how the number of observations also varies by latitude. The number of observations strongly decreases with latitude, especially in March and October, as expected. There are also fewer observations from

the mid May to mid July in the latitudes north of 50°N owing to cloud cover. The number of ground-based observations also varies strongly with fewer observations from May to June owing to instrumental problems at a number of JRSATION sites.

L225-232:
- The source information for JR-STATIONS are available in Data availability section, but how about other ground-based data?

We have added the information for the other ground-based stations as well.

- Did you process/filter these data at all?

The JRSTATION data are given at a frequency of two observations per hour (except for station BRZ for which there are four observations per hour). These observations were averaged to hourly, which we now state in Section 3.1.2. These data were also corrected for a calibration scale offset of 5 ppb with respect to the WMO CH4 X2004A scale (the other observations were already on the WMO CH4 X2004A scale). The in-situ data at station CPA were already provided at hourly resolution. The NOAA flasks at station UUM are sampled in pairs, and these pairs were averaged. For the data obtained from the World Data Centre for Greenhouse Gases (WDCGG), which includes UUM, TER and CPA, quality flags are provided, only data indicated as "invalid" were removed. No other processing was performed.

- For the TROPOMI data, you mentioned that the observation uncertainties were 14-20 ppb. How about for these ground-based observations?

The ground-based observation uncertainties had an interquartile range of 7 to 12 ppb and median value of 8 ppb. We have now added this information to Section 3.1.2.

Section 3.1.3:
- Did all the prior fluxes had estimates for 2020? If not, what did you do? What were the original resolution of the prior fluxes? Did you do any interpolation when original resolution was lower than 0.5°?

The EDGAR-v8 (anthropogenic), LPX-Bern (peatlands, inundated, wet and mineral soils) and the GFED-v4.1s (fire) estimates were all available for the year 2020. The geological and ocean estimates were climatologies. The original resolutions are provided in Table 2. Only the emissions that had lower resolution than 0.5° were the geological ones, which were at 1.0°, these were linearly interpolated to 0.5°.

- Do I understand it correctly that you include ocean emissions, but do not optimise them? How large were the contribution of ocean fluxes to the total fluxes of this domain?

It is correct that we account for the ocean fluxes but do not optimized these. The ocean emissions amounted to 0.5 Tg/y for the domain, and represented 1.6% of the total emissions for the domain (see Table 2).

Section 3.2: I understand that you only optimise total fluxes, but as you find some spatial differences in the flux increments between the TROPOMI and ground-based inversions, can you speculate whether emissions from oil and gas sources have different seasonal patters in the two inversions?

The differences we see between the posterior fluxes from the TROPOMI versus the ground-based observation inversions, is essentially the difference between the prior and posterior fluxes for the ground-based observation inversions, since the posterior fluxes from the TROPOMI inversions remain very close to the prior. The peak in the prior estimate in summer is due to wetland fluxes, and the increase in the summer in the posterior fluxes from the inversion using ground-based observations is likely attributable also to wetland fluxes, as it is unlikely that oil/gas emissions would increase considerably in summer.

L286, L306: Could you add uncertainty estimates as well?

We have added the uncertainties here.

L295-299: Are the number of TROPOMI observations less than those from the ground-based stations? Do you argue that number of the observations was persistent for all months? I suppose flux uncertainties are larger in summer (as a whole domain) due to contribution of wetlands (although it is not so clear from Figure 6)? How would the retrieval biases possibly play a role that were discussed in e.g. Lindqvist et al. (2024)?

There are, as expected, many more TROPOMI observations than there are ground-based ones, however, in terms of the constraint on the fluxes it is not so much the number of observations but how sensitive these are to the fluxes and what uncertainty they have (see also reply to the comment below). The number of TROPOMI observations did vary somewhat from month to month (see new Figure S2 showing the number of observations per flux time step and by latitude).

The prior flux uncertainty is actually constant for each flux time step in the inversion. The prior flux uncertainty is determined as a fraction of the mean flux over all time steps, with a minimum uncertainty equal to approximately the 10th percentile flux value considering all fluxes (averaged in time) over the domain. This gives equal freedom to adjust the fluxes in all seasons.

Concerning retrieval biases, the full-physics retrieval algortihm, RemoTec, which is used to provide the Official ESA retrieval product, is known to be affected by artefacts due to albedo and aerosol size parameter (e.g. Balasus et al. 2024). The WFMD retrieval algorithm is less susceptible to these artefacts, this is also why we chose to use the WFMD retrieval product. Since our posterior fluxes remain very close to the prior ones, we do not see any influence of possible retrieval artefacts on the fluxes from the inversion.

L318-321: Related to questions above, why do you think that the flux were not as well constrained in the TROPOMI inversions? Satellite data are suppose to have good spatial coverage compared to ground-based data, and with much large number of data, it should, in principal, constrain the fluxes better than the ground-based data. But it is not the case here. Is this a general feature or something specific to high northern latitudes?

For the constraint on the fluxes, it is not only how many observations there are, but how sensitive each observation is to the fluxes and what uncertainty they have.

Ground-based observations are made to the largest extent within the boundary layer (possible exception is high mountain stations, which may sometimes, e.g. at nighttime, be above the boundary layer) and thus are more sensitive to surface fluxes. Satellite observations, on the other hand, have varying sensitivity to different layers of the atmosphere and are not as sensitive to the fluxes at the surface.

Furthermore, the ground-based observations are made at much higher accuracy and precision than the satellite observations, which again means these have a stronger constraint on the fluxes. For instance, the median uncertainty in the observation space for the ground-based observations was 8 ppb, whereas it was 16 ppb for the TROPOMI observations, and the model-observation differences are weighted by the inverse of the square of the observation uncertainties.

Another contributing factor for the poorer constraint from the satellite observations is that the satellite SRRs are always smeared out over large regions, whereas the SRRs for ground-based observations can be much more focussed over a small region. The latter leads to larger increments over the background, at least potentially (i.e., if there are sources in that region). This has two effects:
  i)    Larger deviations from the background make the inversion less sensitive to the exact value of the background, and less information is used to optimize the background.
  ii)   Since the cost function (in the inversion) includes the quadratic difference between observation and modelled mixing ratio, a few large differences have more impact than a large number of small differences. This means that the ground-based observations (with more focused SRRs) have more impact.

**Technical comments**

Introduction: Please add information about focus/simulation years

We have now added this information to the last paragraph of the introduction.

L67: Please add references to FLEXPART.

We have now added the reference for FLEXPART v10.4 which is the version used in this study.

Equations: Please use bold fonts for vectors and matrices.

We have corrected all our equations using bold lower case letters for vectors and bold upper case letters from matrices.

L210: …ERA5 at 0.5° x 0.5° and… ?

Yes, we mean 0.5° x 0.5° and include the second 0.5° in the text.

Table 1. What are the altitudes here? Elevation of the site or height from which FLEXPART trajectories were calculated?

These are the sample air intake heights as stated by the data providers.

Figure 2: Are these of super-observations? Please clarify.

These are the super-observations averaged over the month onto a regular grid of 0.25 x 0.25 degrees. We now specify this in the caption.

Figure 4:
- You could perhaps consider adding number of observations here?
- Could you also consider adding ranges?

Fig.4 is the figure showing the area-weighted mean XCH4 over time from the observations and the prior and posterior model. We are not sure where we would add the number of observations here. In any case, we show the number of observations in now in the new Fig. S2.

We are not sure what ranges the reviewer is referring to. If the reviewer means to add the uncertainties for the observations we can do that (i.e., the area-weighted mean uncertainties). As for the modelled XCH4, we do not calculate uncertainties for these, only we assign an uncertainty in the observation space for the inversion to account for the model representation uncertainty and the measurement uncertainty. This, however, is different from the uncertainty on the modelled XCH4 values which necessarily depend on the uncertainty in the fluxes as well as in the model representation.

Please consider combining Figures 5 and 8, and Figures 7 and 9. It would be easy to compare between TROPOMI-based and ground-based inversions that way.

We have combined Figures 5 and 8 showing the prior and posterior fluxes and flux increments from all inversions. However, we prefer not to combine Figures 7 and 9, because we want to keep the focus of Fig. 7 on the comparison of the uncertainty reduction for TROPOMI for all months versus for spring and for summer. Also, considering the formatting it will be akward because it is not possible to fit 4 panels horizontally, and having 3 panels horizontally (i.e. those for TROPOMI) and one panel below would look strange.

**References**

Lindqvist, H., et al.: Evaluation of Sentinel-5P TROPOMI Methane Observations at Northern High Latitudes, Remote Sensing, 16, 2979, https://doi.org/10.3390/rs16162979, 2024.

---

## Referee Report (RR1)

**Efficient use of a Lagrangian Particle Dispersion Model for atmospheric inversions using satellite observations of column mixing ratios**

egusphere-2025-147
Author(s): Rona L. Thompson et al.
MS type: Research article
Iteration: Revised submission

I thank authors for revising the manuscript by taking into account of my comments. But, I noticed that the authors' responses to my comments differ in format from those addressed to Reviewers 1 and 2. While their replies are structured as point-by-point responses with reviewer comments quoted in blue, mine are listed by number without the original comments included. This formatting made it difficult to track and cross-reference of my original concerns. For future revisions, I suggest maintaining a consistent structure that includes the reviewer's comments, as this facilitates the process for both reviewers and the editor.

Even though the manuscript improved significantly from the previous version, not all of the concerns I raised during the initial review have been fully addressed. While the revised methods section is overall clearer, several aspects of the FLEXPART implementation remain vague. I appreciate that the authors corrected and rewrote Equations (6) to (11), as the original formulations contained significant errors. However, it is still unclear whether these errors were confined to the manuscript or if they were also present in the model implementation. I wonder whether the discrepancies arose from a miscommunication between the code developer and the person who wrote the article. If that is the case, it raises the concern that the same misinterpretation may have affected the code implementation itself. I hope this is not the case. Nonetheless, since no new simulations have been provided in this revision, it remains uncertain whether the code implementation is consistent with the corrected equations.

A key unresolved concern in the methods section is with the description of the affine transformation applied to particle coordinates. In my initial review, I requested a clear mathematical description of this transformation. In the revised manuscript, the authors describe the process only qualitatively and refer readers to the Fortran source code in a public repository. This is not an adequate substitute for proper documentation in the manuscript or supplementary material. A clear and explicit mathematical description of the transformation is essential for transparency and reproducibility, particularly because this step directly affects the spatial accuracy of the satellite pixel representation.

Regarding the results and discussion, the authors state that they performed sensitivity tests but chose not to include them, as the paper's emphasis is on methodology. However, if the primary contribution of the manuscript is methodological, it becomes even more important to describe all technical steps, including the transformation, particle release setup, and super-observations procedure, rigorously. Providing this level of detail is critical to ensure that the approach can be reproduced and built upon by other researchers.

To be clear, my goal is not to obstruct publication. I recognize that the manuscript presents valuable developments and has strong potential. However, given that the stated focus is methodological, I believe the technical content should be presented with greater clarity and rigor to meet the standards of transparency and reproducibility expected in the field.

Further comments are below based on the response of the authors:

**Comment 1:**

I recommend adding a brief statement to the main text, perhaps in the Introduction, as there is no dedicated discussion section, to clarify how the developments presented in this study relate to FLEXPART v11. Specifically, while FLEXPART v11 introduces the option for custom particle initialization, it does not currently support simulation of total column averages from satellite retrievals. In contrast, the work presented here, based on FLEXPART v10.4, provides an operational method for calculating total column source-receptor relationships (SRRs) from satellite observations.

It would also be valuable to note that the developments described in this paper are planned for future integration into FLEXPART v11. Including this clarification will help readers and potential users select the appropriate FLEXPART version for their applications and will underscore the significance of the methodological contribution presented here.

**Comment 3:**

In my initial review, I raised concerns regarding the formulation of Equation (6), which originally appeared in the manuscript as:

$$H_{ijk}^{col} \sum_{n=1}^{N} a_n w_n \frac{t}{m_n \rho_{ijk}} \sum_{p=1}^{J_{i,jk,n}} m_p$$

At that time, I pointed out that the variable t should represent the residence time of particles within the surface layer as defined in Seibert and Frank (2004, Eq. 8) and Wu et al. (2018, Eq. 4), rather than a fixed sampling duration. I now see that the authors have updated Equation (6) as follows:

$$H_{in} = \frac{1}{P_n} \sum_{p=1} \frac{l_{pin} \Delta t_{pin}}{\rho_i}$$

While I appreciate the authors' revision of Equation (6) and subsequent equations and their explanation of the transmission function l_pin, I suggest improving the clarity of the description. The current phrasing : l_pin "represents the fraction of the mass remaining in the particle" could be misinterpreted by readers, particularly those less familiar with Lagrangian particle dispersion models operating in backward mode. In this context, particles are tracers and do not carry real mass. Rather, the transmission function serves as a scaling factor to account for atmospheric processes (e.g., chemical losses), modifying each particle's residence time contribution to the SRR.

**Comment 5:**

In my initial review, I suggested the authors to provide the explicit form of the affine transformation applied to particle coordinates (i.e., the mathematical equations used). However, the revised manuscript still describes the method only in qualitative terms, and the authors refer readers to the subroutine releaseparticles_satellite.f90 in the public Fortran code base to understand the implementation.

While I appreciate that the code is openly available, I would like to reiterate that it is neither feasible nor expected for a voluntary reviewer to examine and interpret a large Fortran code base to assess a methodological detail that could, and should, be documented clearly in the manuscript or its supplementary material. Given that the affine transformation plays a critical role in accurately representing satellite pixel geometry, I strongly recommend that the authors include the mathematical formulation of this transformation for transparency and reproducibility.

**Comment 6:**

It would significantly improve the manuscript if the authors provided the explicit equations used to perform the super-observation (super-orbiting) processing of TROPOMI data. Currently, this procedure is only described qualitatively, which limits the reproducibility of the method and may confuse readers unfamiliar with the approach.

**Comment 8:**

Indeed  the reduced-chi-square criterion can be ambiguous and alone is not a sufficient criterion for assessing the appropriateness of the uncertainties. However, as stated by Chevallier et al 2007, in an idealized system (OSSE experiment), the cost function $J(x)$ converges toward N (number of observations). Hopefully, the author made this test with TROPOMI before attempting to run real case inversion.

The reduced chi-square value of 4.86 for the ground-based observations suggests that the observation or model errors may be underestimated, or that there is unaccounted model–data mismatch. It would be helpful if the authors clarified how the observation uncertainty was defined for the in situ network. particularly whether representativeness and model transport errors were included, and at what magnitude.

I made this comment because, the authors mention that *'the satellite observations are more uncertain compared to the ground-based observations (the median uncertainty for XCH4 in our study was 16 ppb compared to 8 ppb for the ground-based observations) and the model-observation errors are weighted by the inverse square of the uncertainty'*. It could be possible that the uncertainties of ground based observations in combination with the model errors are larger that 8 ppb, hence the reduced chi-square value is super high (4.86).

**Comment 9:**

The authors note that they performed several sensitivity tests but chose not to include them in the manuscript, as the focus of the paper is "primarily on the methodology of using satellite observations in an inversion framework based on a Lagrangian transport model." However, if the manuscript is intended to be primarily methodological, then the methodological components should be described in greater detail.

Given the concise length and the technical nature of the work, specifically, the adaptation of FLEXPART v10.4 for use with satellite-based retrievals, the manuscript should contain the formal presentation of the algorithmic steps, including the mathematical formulation of transformations and other implementation choices, which are currently either omitted or described only qualitatively.

---

## Author Response (AR2)

Referee #1

• Figure 4: with ranges, I meant range of XCH4 values within a month and the domain. Could be standard deviations or quantiles. I suppose you can possibly add for both observations and modelled values.

Thanks for the clarification. We have added the standard deviations to both the observed and modelled mixing ratios in Fig. 4.

• Following the comments by the reviewer 3, please shortly add reasons for the choice of the FLEXPART version in Introduction or Method section and your plan to implement the code in FLEXPARTv11 in Conclusion.

We implemented this methodology in FLEXPART v10.4 simply because v11 was not ready (also with no outlook of when it would be ready) when we started this work. (Note that the FLEXPART-v11 paper was only published 30 October 2024) . Since the methodology is general (it could be implemented in other LPDMs) we do not want to focus too much on FLEXPART in the paper. We, however, include a brief description of how the methodology can be used with FLEXPART-v11 in the Supplement.

**Reply to Reviewer 2**

We thank the reviewer for these comments and reply to them point-by-point below. Please note our responses are indicated in blue font.

I thank authors for revising the manuscript by taking into account of my comments. But, I noticed that the authors' responses to my comments differ in format from those addressed to Reviewers 1 and 2. While their replies are structured as point-by-point responses with reviewer comments quoted in blue, mine are listed by number without the original comments included. This formatting made it difficult to track and cross-reference of my original concerns. For future revisions, I suggest maintaining a consistent structure that includes the reviewer's comments, as this facilitates the process for both reviewers and the editor.

The different format was simply because your review comments were submitted in a PDF format that was not recognizable as text. This meant that we could not simply copy the text from the PDF into a new document and include our replies. This time round a work around was found by scanning the PDF and converting it to text. Despite the different format in the previous response, please note that we did take care to respond to all comments and our responses were numbered according to the comment they addressed.

Even though the manuscript improved significantly from the previous version, not all of the concerns I raised during the initial review have been fully addressed. While the revised methods section is overall clearer, several aspects of the FLEXPART implementation remain vague. I appreciate that the authors corrected and rewrote Equations (6) to (11), as the original formulations contained significant errors. However, it is still unclear whether these errors were confined to the manuscript or if they were also present in the model implementation. I wonder whether the discrepancies arose from a miscommunication between the code developer and the person who wrote the article. If that is the case, it raises the concern that the same misinterpretation may have affected the code implementation itself. I hope this is not the case. Nonetheless, since no new simulations have been provided in this revision, it remains uncertain whether the code implementation is consistent with the corrected equations.

The same person (Rona Thompson) wrote the code and the manuscript, so there is no miscommunication. In the first version of the manuscript, we provided simplified versions of the equations, which we thought it would make it easier for the readers to follow, e.g., the transmission function was omitted in Eq. 6 as this term is does not significantly differ from one. The correct equations were implemented in the code and are consistent with the revised version of the manuscript.

A key unresolved concern in the methods section is with the description of the affine transformation applied to particle coordinates. In my initial review, I requested a clear mathematical description of this transformation. In the revised manuscript, the authors describe the process only qualitatively and refer readers to the Fortran source code in a public repository. This is not an adequate substitute for proper documentation in the manuscript or supplementary material. A clear and explicit mathematical description of the transformation is essential for transparency and reproducibility, particularly

because this step directly affects the spatial accuracy of the satellite pixel representation.

We did not previously include the explicit calculations for the affine transformation because this is an algorithm and not just a few equations. We now include the code excerpt for this with explanations in the Supplement.

Regarding the results and discussion, the authors state that they performed sensitivity tests but chose not to include them, as the paper's emphasis is on methodology. However, if the primary contribution of the manuscript is methodological, it becomes even more important to describe all technical steps, including the transformation, particle release setup, and super-observations procedure, rigorously. Providing this level of detail is critical to ensure that the approach can be reproduced and built upon by other researchers.

The main focus on the paper is the methodology for modeling column observations, such as those from satellites, in an efficient way using a Lagrangian particle dispersion model. The case study is given to demonstrate the application of the methodology but is not intended to be the focus of the manuscript. We have, however, included more technical details in the paper, specifically, we now include the excerpt of code for the affine transformation in the Supplement.

To be clear, my goal is not to obstruct publication. I recognize that the manuscript presents valuable developments and has strong potential. However, given that the stated focus is methodological, I believe the technical content should be presented with greater clarity and rigor to meet the standards of transparency and reproducibility expected in the field.

We have now added further explanations on the methodology in the Supplement. We chose to put it there as we feel that having these extra explanations in the main part of the manuscript would make the paper too long and distract from the main focus.

Further comments are below based on the response of the authors:

**Comment 1:**

I recommend adding a brief statement to the main text, perhaps in the Introduction, as there is no dedicated discussion section, to clarify how the developments presented in this study relate to FLEXPART v11. Specifically, while FLEXPART v11 introduces the option for custom particle initialization, it does not currently support simulation of total column averages from satellite retrievals. In contrast, the work presented here, based on FLEXPART v10.4, provides an operational method for calculating total column source-receptor relationships (SRRs) from satellite observations.

It would also be valuable to note that the developments described in this paper are planned for future integration into FLEXPART vII. Including this clarification will help readers and potential users select the appropriate FLEXPART version

for their applications and will underscore the significance of the methodological contribution presented here.

We have now included a short section in the Supplement about how these developments could be used in conjunction with FLEXPART-v11. In FLEXPART-v11, the particle initial positions can be specified by reading in a NetCDF file (this was not an option with previous FLEXPART versions). The developments described in this paper can be used to create such a NetCDF file specifying the particle positions for the releases. This would involve the steps from the pre-processor developed for FLEXPARTv10.4 and FLEXINVERT, which reads the satellite observations (L2 data) and performs optional averaging of retrievals, and then use the code "releaseparticles_satellite.f90" written for FLEXPART-v10.4, which performs the affine transformation and calculates the release positions for all particles.

We think this description does not belong in the main part of the paper because the focus of the paper is not about the development of FLEXPART. Instead, the focus is on the methodology of calculating source receptor relationships for satellite observations, which is general in the sense that it could be implemented in any Lagrangian transport model.

**Comment 3:**

In my initial review I raised concerns regarding the formulation of Equation (6), which originally appeared in the manuscript as:

$$H_{ijk}^{col} \sum_{n=1}^{N} a_n w_n \frac{t}{m_n \rho_{ijk}} \sum_{p=1}^{J_{i,jk,n}} m_p$$

At that time, I pointed out that the variable t should represent the residence time of particles within the surface layer as defined in Seibert and Frank (2004, Eq. 8) and Wu et al. (2018, Eq. 4), rather than a fixed sampling duration. I now see that the authors have updated Equation (6) as follows:

$$H_{in} = \frac{1}{P_n} \sum_{p=1} \frac{l_{pin} \Delta t_{pin}}{\rho_i}$$

While I appreciate the authors' revision of Equation (6) and subsequent equations and their explanation of the transmission function l_pin, I suggest improving the clarity of the description. The current phrasing : l_pin "represents the fraction of the mass remaining in the particle" could be misinterpreted by readers, particularly those less familiar with Lagrangian particle dispersion models operating in backward mode. In this context, particles are tracers and do not carry real mass. Rather, the transmission function serves as a scaling factor to account for atmospheric processes (e.g., chemical lasses), modifying each particle's residence time contribution to the SRR.

In FLEXPART particles are assigned an initial (arbitrary) mass, which can change e.g. due to chemical loss, and is the implementation of the transmission function.

**Comment 5:**

In my initial review, I suggested the authors to provide the explicit form of the affine transformation applied to particle coordinates (i.e., the mathematical equations used). However, the revised manuscript still describes the method only in qualitative terms, and the authors refer readers to the subroutine releaseparticles_satellite.190 in the public Fortran code base to understand the implementation.

While I appreciate that the code is openly available, I would like to reiterate that it is neither feasible nor expected for a voluntary reviewer to examine and interpret a large Fortran code base to assess a methodological detail that could, and should, be documented clearly in the manuscript or its supplementary material. Given that the affine transformation plays a critical role in accurately representing satellite pixel geometry, I strongly recommend that the authors include the mathematical formulation of this transformation for transparency and reproducibility.

We now include excerpts of the code for the affine transformation (with explanation) in the Supplement.

**Comment 6:**

It would significantly improve the manuscript if the authors provided the explicit equations used to perform the super-observation (super-orbiting) processing of TROPOMI data. Currently, this procedure is only described qualitatively, which limits the reproducibility of the method and may confuse readers unfamiliar with the approach.

The code that performs the calculation of the optimal grid for the retrievals (note that it is not specific to TROPOMI) and the averaging of the retrievals to this grid is over 400 lines. We have never seen a paper in Atmos. Chem. Phys. that includes entire algorithms or code. We describe the principle of how the algorithm calculates the optimal grid in Section 2.2, and we give the equations for the averaging of the retrievals to super-observations in Equations 12 to 14. The code is open source and anyone who is interested in how the algorithm is implemented can download the code from the Git repository.

**Comment 8:**

Indeed the reduced-chi-square criterion can be ambiguous and alone is not a sufficient criterion for assessing the appropriateness of the uncertainties. However, as stated by Chevallier et al 2007, in an idealized system (OSSE experiment), the cost function J(x) converges toward N (number of observations). Hopefully, the author made this test with TROPOMI before attempting to run real case inversion.

We did run synthetic data experiments to check the validity of the code and settings. For the TROPOMI inversions the reduced chi-square value is 1.08 and is very close to the "ideal" value of 1.0.

The reduced chi-square value of 4.86 for the ground-based observations suggests that the observation or model errors may be underestimated, or that there is unaccounted model-data mismatch. It would be helpful if the authors clarified how the observation uncertainty was defined for the in situ network. particularly whether

representativeness and model transport errors were included, and at what magnitude.

We agree that the reduced-chi square value is high for the ground-based observation inversion. The observation space uncertainties include the measurement uncertainty, a proxy for the transport uncertainty was taken as the standard deviation of the measurement in one hour for continuous measurements and a set value of 5 ppbv for flask measurements. For continuous observations, if the standard deviation was < 5ppb the minimum estimate of 5 ppbv for the transport uncertainty was used. In addition, we calculated an estimate for the background uncertainty.

In this revision, we re-ran the inversion increasing the observation space uncertainties by 5 ppbv with respect to their former values. The reduced-chi square value decreased to 2.16. Also, the median of the uncertainty is now 11.9 ppbv, while the median of the absolute posterior model – observation error is 10.6 ppbv, indicating a reasonable estimation of the observation space uncertainty.

I made this comment because, the authors mention that *'the satellite observations are more uncertain compared to the ground-based observations (the median uncertainty for XCH4 in our study was 16 ppb compared to 8 ppb for the ground-based observations) and the model-observation errors are weighted by the inverse square of the uncertainty'*. It could be possible that the uncertainties of ground based observations in combination with the model errors are larger that 8 ppb, hence the reduced chi-square value is super high (4.86).

We have revised our observation space uncertainties upwards resulting now in a median of 12 ppb. The posterior fluxes from the inversion with this larger observation space uncertainty are very similar to those from the original inversion. The mean posterior source over the domain is now 34.6 compared to the previous value of 33.9 Tg/y. The posterior uncertainties are also very similar to the original inversion. We have updated figures 5, 6 and 8 and figure S3b and the text for the new inversion results and uncertainty estimates.

The ground-based observation space uncertainties still remain lower than those for the satellite observations, 12 ppb versus 16 ppb for the median uncertainty, respectively. Moreover, in our discussion we give two principal reasons for why the inversions using TROPOMI have a lower uncertainty reduction and why the posterior fluxes remain very close to the prior ones. The first is due to the higher uncertainty in the TROPOMI observations compared to the ground-based ones (which still holds after the upward revised uncertainties in the ground-based observation inversion). The second, and even more importantly, is the fact that each single retrieval (or TROPOMI observation) is much less sensitive to the fluxes in the domain leading to smaller model-observation differences. Since the cost function depends on the square of the model-observation differences, a few large differences have more influence on the cost than many small ones.

**Comment 9:**

The authors note that they performed several sensitivity tests but chose not to include them in the manuscript, as the focus of the paper is "primarily on the methodology of using satellite observations in an inversion framework based on a Lagrangian

transcript model." However, if the manuscript is intended to be primarily methodological, then the methodological components should be described in greater detail.

We describe the methodology for the calculation of the source receptor relationships for satellite observations, and the methodology for the averaging of the retrievals, in detail. Furthermore, we discuss the key factors that the methodology is sensitive to. Specifically, we discuss the sensitivity to the initial mixing ratios, which are used to calculate the background column mixing ratios, and is the most important factor for determining the results. We present results using different initial mixing ratios and discuss how this sensitivity can be resolved by optimizing scalars of the initial mixing ratios (see section 3.2.2). We also discuss the choice of uncertainty for the scalars on the initial mixing ratios, and how these uncertainties were increased based on the results of sensitivity tests (see L226). We reiterate, that the focus of the paper is the methodology for the calculation of the source receptor relationships for satellite observations, and that the case study is to demonstrate the application of the methodology but is not intended to be the focus itself.

Given the concise length and the technical nature of the work, specifically, the adaptation of FLEXPART vl0.4 for use with satellite-based retrievals, the manuscript should contain the formal presentation of the algorithmic steps, including the mathematical formulation of transformations and other implementation choices, which are currently either omitted or described only qualitatively.

We include the key equations for the methodology of calculating source-receptor relationships using an LPDM in the manuscript (Eq. 1 to 11), and the key equations for the averaging of the retrievals (Eq 12 to 14). We have also in this revision included the algorithm for the affine transformation in the Supplement. It is not normal practice in Atmos. Chem. Phys. to include entire algorithms and code in papers. Specifically, the algorithm for the optimal averaging for super observations is over 400 lines long, and those readers who are interested in exactly how this is implemented can access the code from the open Git repository (details are in the section "Data Availability") and they are also welcome to contact the corresponding author (Rona Thompson).